

# Computational perspectives revealed prospective vaccine candidates from five structural proteins of novel SARS corona virus 2019 (SARS-CoV-2)

Rajesh Anand*, Subham Biswal*, Renu Bhatt and Bhupendra N. Tiwary

Department of Biotechnology, Guru Ghasidas Vishwavidyalaya, (A Central University), Bilaspur, Chhattisgarh, India
* These authors contributed equally to this work.

## ABSTRACT

**Background:** The present pandemic COVID-19 is caused by SARS-CoV-2, a single-stranded positive-sense RNA virus from the *Coronaviridae* family. Due to a lack of antiviral drugs, vaccines against the virus are urgently required.

**Methods:** In this study, validated computational approaches were used to identify peptide-based epitopes from six structural proteins having antigenic properties. The Net-CTL 1.2 tool was used for the prediction of CD8[+] T-cell epitopes, while the robust tools Bepi-Pred 2 and LBtope was employed for the identification of linear B-cell epitopes. Docking studies of the identified epitopes were performed using HADDOCK 2.4 and the structures were visualized by Discovery Studio and LigPlot[+]. Antigenicity, immunogenicity, conservancy, population coverage and allergenicity of the predicted epitopes were determined by the bioinformatics tools like VaxiJen v2.0 server, the Immune Epitope Database tools and AllerTOP v.2.0, AllergenFP 1.0 and ElliPro.

**Results:** The predicted T cell and linear B-cell epitopes were considered as prime vaccine targets in case they passed the requisite parameters like antigenicity, immunogenicity, conservancy, non-allergenicity and broad range of population coverage. Among the predicted CD8+ T cell epitopes, potential vaccine targets from surface glycoprotein were; YQPYRVVVL, PYRVVVLSF, GVYFASTEK, QLTPTWRVY, and those from ORF3a protein were LKKRWQLAL, HVTFFIYNK. Similarly, RFLYIIKLI, LTWICLLQF from membrane protein and three epitopes viz; SPRWYFYYL, TWLTYTGAI, KTFPPTEPK from nucleocapsid phosphoprotein were the superior vaccine targets observed in our study. The negative values of HADDOCK and Z scores obtained for the best cluster indicated the potential of the epitopes as suitable vaccine candidates. Analysis of the 3D and 2D interaction diagrams of best cluster produced by HADDOCK 2.4 displayed the binding interaction of leading T cell epitopes within the MHC-1 peptide binding clefts. On the other hand, among linear B cell epitopes the majority of potential vaccine targets were from nucleocapsid protein, viz; [59-]HGKEDLKFPRGQGVPINTNSS PDDQIGYYRRATRRIRGGDGKMKDLS[-105], [227-]LNQLE SKMSGKGQQQQGQT VTKKSAAEASKKPRQKRTATK[-266], [3-]DNGPQNQRNAPRITFGGP[-20], [29-]GERSG ARSKQRRPQGL[-45]. Two other prime vaccine targets, [370-]NSASFSTFKCYGVSPTK LNDLCFTNV[-395] and [260-]AGAAAYYVGYLQPRT[-274] were identified in the spike

Corresponding author
Rajesh Anand, rajesh9in@gmail.com

protein. The potential B-cell conformational epitopes were predicted on the basis of a higher protrusion index indicating greater solvent accessibility. These conformational epitopes were of various lengths and belonged to spike, ORF3a, membrane and nucleocapsid proteins.

**Conclusions:** Taken together, eleven T cell epitopes, seven B cell linear epitopes and ten B cell conformational epitopes were identified from five structural proteins of SARS-CoV-2 using advanced computational tools. These potential vaccine candidates may provide important timely directives for an effective vaccine against SARS-CoV-2.

## INTRODUCTION

Globally the present dreaded pandemic of coronavirus disease 2019 (COVID-19) has resulted in the deaths of more than 445,000 humans (*World Health Organization, 2019*). The causative agent of the disease has been severe acute respiratory syndrome coronavirus 2 (SARS-CoV-2) (*World Health Organization, 2019*). The family Coronaviridae consists of a large group of viruses known as coronaviruses (CoVs). The corona viruses were thought to be harmless respiratory human pathogens due to (i) harmless mild infections and (ii) the limited number of the circulating viruses in humans (*Song et al., 2019*). However, the emergence of a series of three severe and fatal diseases caused by corona virus changed the concept. The first instance was severe acute respiratory syndrome (SARS) in November 2002–February 2003 in China and the second was Middle East Respiratory Syndrome (MERS) in June 2012 in Saudi Arabia (*De Wit et al., 2016*). The most recent cases of fatal disease outbreaks caused by corona virus occurred in December 2019, in Wuhan, Hubei, China. These consecutive viral outbreaks also indicate the threat of cross-species transmission of these viruses leading to severe infectious outbreak in humans that should be considered seriously (*Menachery et al., 2015*). Therefore, the threats of CoVs should not be undermined and the research on the life cycle and host-virus interactions should be advanced in order to develop treatments and vaccines against these viruses. The scientific and clinical investigations demonstrated that SARS-CoV and MERS-CoV share remarkable features that lead to preferential viral replication in the lower respiratory tract and viral immunopathology. The recent investigations on the clinical, laboratory, radiological and epidemiological characteristics and outcomes of treatments in patients demonstrated that the severe respiratory illness similar to SARS-CoV was due to SARS-CoV-2 (COVID-19) (*Huang et al., 2020*). Although the early investigations patterns suggested that the COVID-19 virus could cause severe illness in some patients, with limited transmission among people, up-to-date epidemiological data strongly favors the statement that the new virus has evolved/adapted more efficiently for transmission among humans. The genome sequences of COVID-19 viruses obtained from patients indicated that they share 79.5% sequence identity to SARS-CoV (*Zhou et al., 2020*) and a 96% identity to bat corona virus at the whole genome level. The phylogenetic

studies of corona viruses obtained from different organisms indicated that COVID-19 could have originated from Chinese horseshoe bats; however, the vehicle which led to the transmission to host has not yet been identified (*Dong et al., 2020*). The COVID-19 virus was bannered as a novel type of corona virus from bat due to a high degree of variation from the human SARS virus (*Shereen et al., 2020*; *Andersen et al., 2020*). Altogether seven member of the family of CoVs infect humans, the COVID-19 is the newest of all. The SARS-CoV and hCoV-NL63 utilizes human angiotensin converting enzyme 2 (ACE2) for virus entry (*Hofmann et al., 2005*; *Li et al., 2003*; *Wu et al., 2009*). Recently scientists have identified that both the SARS-corona virus and the COVID-19 virus enter host cells through an endosomal pathway involving the same entry receptor ACE2 (*Zhou et al., 2020*; *Letko, Marzi & Munster, 2020*). The entry process of corona viruses is facilitated by the surface-located spike glycoprotein (*Lu, Wang & Gao, 2015*). The spike protein can be divided into the S1 and S2 subunits, which are utilized as receptor recognition and membrane fusion molecules, respectively (*Lai, Perlman & Anderson, 2007*). S1 Both the N-terminal domain (NTD) and a C-terminal domain (CTD) of S1 unit can function as a receptor-binding entity or receptor binding domain (RBD) (*Li et al., 2005*; *Lu et al., 2013*; *Taguchi & Hirai-Yuki, 2012*). Recently, the S1 CTD (SARS-CoV-2-CTD) has been identified as the prime region in SARS-CoV-2 that interacts with the ACE2 receptor (*Wang et al., 2020*). The crystal structure of SARS-CoV2-CTD in complex with human ACE2, exhibited bindings similar to that observed for the SARS-CoV-RBD. It has been further identified that SARS-CoV-2-CTD forms more atomic interactions with human ACE2 than SARS-RBD, resulting in higher affinity for receptor binding (*Shang et al., 2020*; *Wang et al., 2020*).

On the basis of the genetic properties, the *Coronaviridae* family can be divided in to four genera, including genus Alpha corona virus, genus Beta corona virus, genus Gamma corona virus, and genus Delta corona virus. Among the RNA viruses, the corona virus has the largest genome (ranging from 26 to 32 kb) with particle size of viruses being about 125 nm in diameter (*Ji et al., 2020*). CoVs possess a composite genome expression strategy as numerous CoV proteins expressed in the infected cell contribute to the corona virus-host interactions. These strategies include (i) associations with the host cell to create a favorable environment for CoV replication, (ii) modification of the host gene expression and nullifying the antiviral defenses of host. The CoV-host interplay is thus key to pathogenesis of virus (*De Wilde et al., 2018*). Two-thirds of the CoV genome belongs to genes for non-structural proteins. Amid the structural proteins, spike (S), envelope (E), membrane (M) and nucleocapsid (N) can be considered important in terms of vaccine potential. The viral membrane has S, E and M proteins. The spike protein is a surface-located trimeric glycoprotein and actively plays a role in viral ingress into host cells, viral infection, and pathogenesis and was contemplated as a prime vaccine and therapeutic target against SARS-CoV and MERS-CoV. The membrane and envelope proteins are required in viral assemblage, whereas the nucleocapsid protein is involved for assembly of RNA genome (*Song et al., 2019*).

Although CoVs share numerous resemblances, they also have genetically evolved significantly and finding the potential targets for vaccines and antiviral drugs against

COVID-19 should exploit the structural similarities between SARS-CoV and the COVID-19 virus, and focus on proteins that are highly conserved across multiple CoVs. In this work, all the structural proteins were selected for finding the epitopes for designing the vaccine against CoVID-19 using validated in silico approaches.

# MATERIALS AND METHODS

The overall procedures used in the present study for epitope-based vaccine design and physicochemical property prediction have been depicted in the form of flow chart (Fig. 1).

## Retrieval of the protein sequence

The protein sequences of SARS-CoV-2 were retrieved from the Virus Pathogen Database and Analysis Resource (ViPR) (http://www.viprbrc.org/) in FASTA format. ViPR database helps in fetching the sequence from both GenBank and UniProtKB in the FASTA format (*Pickett et al., 2012*). The nonstructural proteins of SARS-CoV-2 were removed from the complete proteome of SARS-CoV-2.

## Similarity search and selection of protein for epitope prediction

The sequences of potential structural proteins (surface glycoprotein, orf3a protein, envelope protein, membrane glycoprotein, nucleocapsid phosphoprotein, orf6 protein) were searched for the similarity using the BLAST tool (https://blast.ncbi.nlm.nih.gov/Blast.cgi?PAGE=Proteins). Only those hits were selected for comparison of identities which showed 100% query cover. The BLAST results of surface glycoprotein (Accession No. QHQ82464) exhibited more than 99.84% identity (100 hits) and those of orf3a (Accession No. QHQ82465) showed identities of more than 99.27% (74 hits). Next the envelope protein (Accession No. QHQ82466) resulted in 14 hits with the identity range between 100% and 94.67% while the BLAST using the membrane glyco-protein (Accession No. QHQ82467) resulted in about 50 hits with the range of identity between 100% and 93.24%. Similarly the nucleocapsid phosphoprotein (Accession No. QHQ82471) resulted in 100 hits with identity in the range from 100% to 99.28%. The last selected protein from SARS-CoV-2, orf6 (Accession No. QHQ82468) exhibited an identity range between 100% and 95% in more than 40 hits. Since all the BLAST searches performed using the above structural proteins demonstrated the identity between 93% and 100%, a conclusion was drawn that any one representative protein from the six structural proteins could be used for further study. Thus we selected all the six complete sequences of proteins with the above accession numbers for further studies.

## Determination of antigenicity of the SARS-CoV-2 MHC I epitopes

The term antigenicity is the capacity of a molecule to be specifically recognized by the antibodies generated as a result of immune response to the given substance. Proteins generated by divergent or convergent evolution may lack apparent sequence similarity, although they may share structural similarity and biological characteristics (*Petsko & Ringe, 2004*). Antigenicity may be encoded in a sequence in a fine and obscure manner not feasible to direct recognition by sequence alignment. Similarly, the search of novel antigens will be circumvented by their lack of similarity to antigens of known origin (*Doytchinova &*

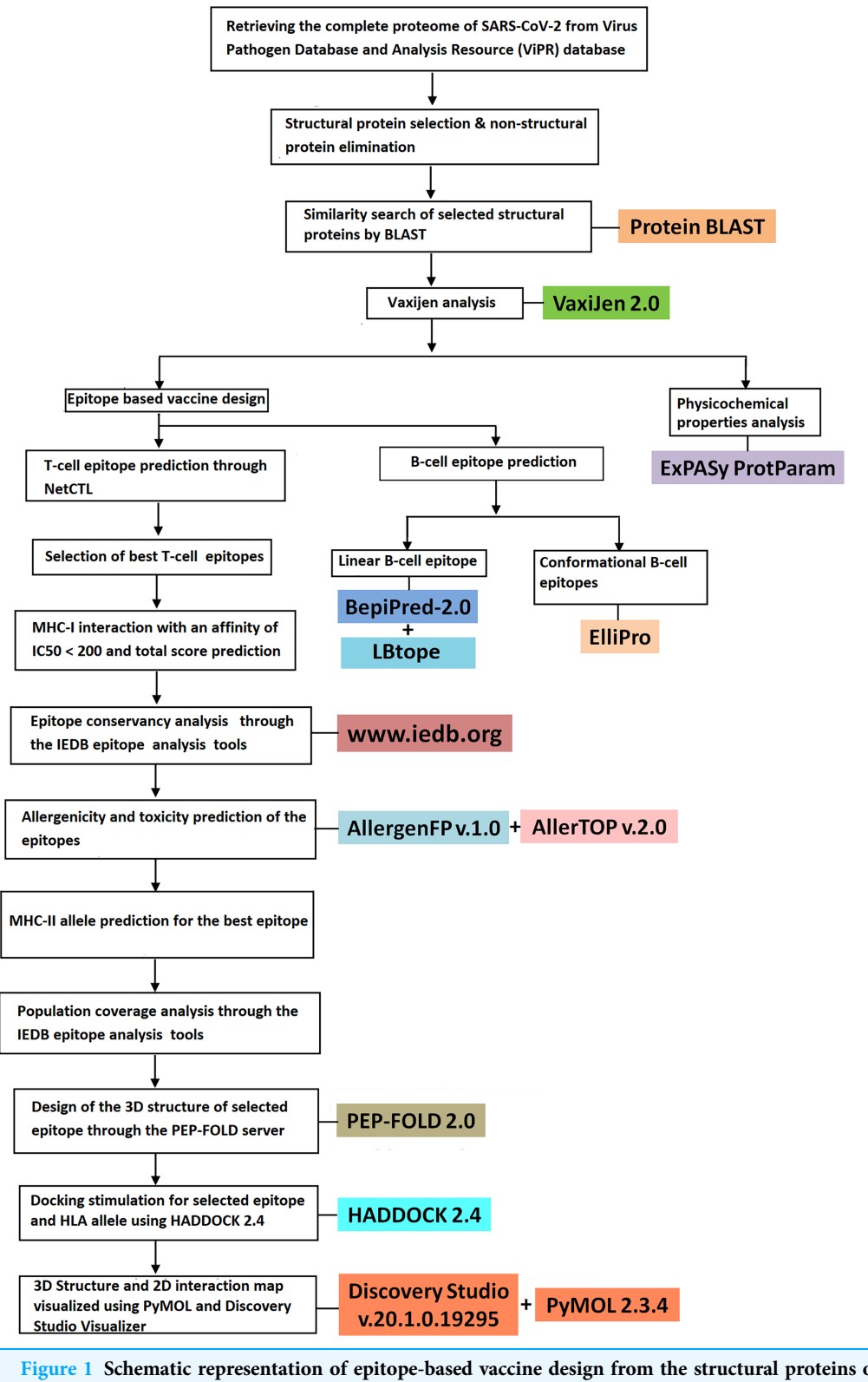

**Figure 1** **Schematic representation of epitope-based vaccine design from the structural proteins of SARS-CoV-2.**                           

*Flower, 2007*). A novel alignment-free approach for antigen prediction, VaxiJen, based on auto cross covariance (ACC) transformation of protein sequences into uniform vectors of principal amino acid properties was developed to control the limitations of alignment based strategies (*Doytchinova & Flower, 2007*). All the structural proteins of SARS-CoV-2 were submitted to the VaxiJen v2.0 server (http://www.ddg-pharmfac.net/vaxijen/VaxiJen/VaxiJen.html) (*Doytchinova & Flower, 2007*) in FASTA format for the determination of antigenicity. A threshold value of 0.4 was considered for determination of antigenicity.

## Prediction and identification of T Cell Epitopes

The T-cell epitopes are usually small peptide fragments of 8–11 amino acids and can elicit specific immune responses. These are important for epitope-based peptide vaccine design (*Patronov & Doytchinova, 2013*). The NetCTL 1.2 server (http://www.cbs.dtu.dk/services/NetCTL/), can be utilized for the prediction of the T cell epitopes in any specified protein. The server can anticipate the epitope for 12 MHC-I super types A1, A2, A3, A24, A26, B7, B8, B27, B39, B44, B58, B62 present on CD8[+] T Cells. The prediction of epitope of CD8[+] T cell is on the basis of interpretations obtained from proteasomal C terminal cleavage, MHC class I binding, and TAP transport efficiency. The artificial neural network (ANN) is used for the prediction of MHC class I binding, proteasomal C terminal cleavage, while weight matrix is employed for the estimation of TAP transport efficiency (*Larsen et al., 2005*, *2007*). In this study the server was used for the prediction of epitopes of all the 12 super types of MHC I and a higher threshold value of 1.25 for epitope prediction was fixed, which has a better sensitivity and specificity of 0.54 and 0.993, respectively. The default parameters set by server for the weight matrix determination for proteasomal C terminal cleavage (0.15) and TAP transport efficiency (0.05) were used.

## Prediction of antigenicity

For the recognition of both frequently and non-frequently occurring MHC-I-binding alleles, the T cell epitopes of SARS-CoV-2 were analyzed by the stabilized matrix base method (SMM) of the Immune Epitope Database (IEDB) analysis tool (http://tools.iedb.org/mhci/) as described earlier (*Peters et al., 2005*; *Lundegaard et al., 2008*). The recognition of MHC-I binding alleles were performed on the parameters (i) the peptide length of epitope was restricted to 9 and (ii) the IC50 of less than 250 nM was selected on the server. Lower IC50 value signifies higher binding. The HLA-binding affinity of the epitopes is quantitatively described in the IC50 nM units. In general, for similar ligands, higher binding affinity of the epitopes with the MHC class I molecule is reflected by the lower IC50 value. Therefore, IC50 values less than 250 nM (IC50 < 250) was selected for ensuring higher binding affinity of the epitopes. The IC50 of the epitopes were determined by the IEDB tool. IEDB being a resourceful server, can also be used for the estimation of processing score, TAP score, proteasomal cleavage, and the MHC-I binding score of the specified epitopes and their respective alleles using the stabilized matrix based method (*Peters et al., 2003*; *Tenzer et al., 2005*). Epitopes were selected based on the highest combined score, but the final selection for further study was made after the prediction of antigenicity by VaxiJen v2.0 server and that of immunogenicity by IEDB server. The combined score is the derived from median

percentile rank of seven alleles and immunogenicity score using following equation Combined percentile rank = (alpha × Immunogenicity model score) + ((1 − alpha) × Median Percentile rank of seven alleles), where alpha is optimized to 0.4 as described earlier (*Dhanda et al., 2018*).

## Epitope immunogenicity and conservancy prediction

Immunogenicity is defined as the ability of a substance/molecule to instigate cellular and humoral immune response (*Ilinskaya & Dobrovolskaia, 2016*). Conservancy may be defined as the fragment of protein sequences that carry the epitope which is considered at or above a specified level of identity (*Bui et al., 2007*). The effective T-cell epitopes are more immunogenic and are considered better than the less immunogenic peptides (*Adhikari, Tayebi & Rahman, 2018*). Therefore, the epitope with better immunogenicity was selected for further evaluation. The immunogenicity prediction tool available on the server http://tools.iedb.org/immunogenicity/ was utilized for the identification of immunogenicity while conservancy was predicted by the tool available on iedb (http://tools.iedb.org/conservancy/) (*Nielsen, Lundegaard & Lund, 2007*; *Calis et al., 2013*). All the epitopes having positive immunogenicity scores (given by IEDB tool) were considered a potential immunogen.

## Determination of population coverage

MHC molecules are exceptionally polymorphic and more than a thousand divergent human MHC (HLA) alleles are recognized. To determine the population coverage a tool is required that can optimally calculate the distribution of humans which will respond to a given group of epitope on the basis of HLA genotypic prevalence and MHC binding and/or T cell restriction data (*Bui et al., 2007*). Population coverage for each identified epitope and their corresponding MHC HLA-binding alleles was determined by the population coverage tool available on IEDB server (http://tools.iedb.org/population/). Here we used the allelic frequency of the interacting HLA alleles for the prediction of the population coverage for the corresponding epitope. In a recent report population coverage of about 64% was reported for an epitope (*Oany, Emran & Jyoti, 2014*). In this study a population coverage of 65% or more was selected.

## Allergenicity and toxicity assessment

The web-based AllerTOP v.2.0 (http://www.ddg-pharmfac.net/AllerTOP/) (*Dimitrov et al., 2014a*) and AllergenFP 1.0 (http://www.ddg-pharmfac.net/AllergenFP/) (*Dimitrov et al., 2014b*) was used to check the allergenicity of our proposed epitope for vaccine development. AllergenFP 1.0 has been established on a novel alignment-free descriptor-based fingerprint technique. An accuracy of 87.9% is observed in the identification of both allergens and nonallergens by AllergenFP 1.0. In contrast, to classify allergens and nonallergens, AllerTOP v. 2.0 has been established on the basis of *k*-nearest neighbors (kNN) method. The web server ToxinPred (http://crdd.osdd.net/raghava/toxinpred/) was implemented to predict toxicity of the peptides (*Gupta et al., 2013*). This strategy was developed the basis of machine learning technique and quantitative matrix utilizing distinctive properties of peptides.

## Prediction of MHC II epitopes

The disadvantage of many bioinformatics methods including Gibbs samplers, Ant colony, Artificial neural networks, Support vector machines, hidden Markov models, and motif search algorithms for predicting MHC class II epitopes is owing to training and evaluation on very limited data sets covering a single or a few different MHC class II alleles (*Nielsen, Lundegaard & Lund, 2007*). On the IEDB database, a large group of quantitative MHC class II peptide-binding data is available (*Toseland et al., 2005*). The data includes the peptide with binding affinities (IC50) for more than 14 HLA/MHC. A novel stabilized matrix method (SMM)-align method (*NetMHCII*) for quantitative predictions of MHC class II binding was developed which utilizes the IEDB MHC class II peptide binding database (*Nielsen, Lundegaard & Lund, 2007*). The SMM-align method attempts to recognize a weight matrix that ideally emulates the measured IC50 values for each peptide in the training group (*Nielsen, Lundegaard & Lund, 2007*). The MHC I epitopes derived from structural proteins were selected for the prediction of MHC-II-binding alleles using the SMM-align method. As per the instruction of the tool, an IC 50 value up to 3000 nM was considered significant.

## Design of the three-dimensional (3D) structure of epitope

In order to be considered as proper vaccine candidate, an epitope need to fulfill all the criteria like antigenicity, immunogenicity, conservancy of epitopes, non-toxicity and it should be non-allergen. Epitope candidates were evaluated on the basis of above parameters and were subjected to the determination of three-dimensional structure using the PEP-FOLD peptide prediction server (http://bioserv.rpbs.univ-paris-diderot.fr/services/PEP-FOLD/) (*Thévenet et al., 2012*; *Shen et al., 2014*). Thus the potential epitopes fulfilling all the above criteria were used for the structure determination. The best model obtained using the server was taken forward for docking analysis.

## Docking analysis

To know the binding interactions between HLA molecules and the predicted epitope, molecular docking simulation was executed using High Ambiguity Driven protein-protein DOCKing (HADDOCK) version: 2.4 (https://bianca.science.uu.nl/haddock2.4/). HADDOCK is an information-driven flexible docking approach for the modeling of biomolecular complexes. Despite continuous advances in the field, the accuracy of ab initio docking-without using any experimental restraints-remains generally low (*Huang, 2015*). Data-driven approaches such as HADDOCK (*Van Zundert et al., 2016*), which integrate information derived from biochemical, biophysical or bioinformatics methods to enhance sampling, scoring or both (*Rodrigues & Bonvin, 2014*), perform remarkably better. The main attribute of HADDOCK is the Ambiguous Interaction Restraints or AIRs. These permit the conversion of raw data including mutagenesis experiments or NMR chemical shift perturbation into distance restraints which are integrated in the energy functions. These energy functions are used in calculations. In the docking protocol of HADDOCK, molecules pass through varying degrees of flexibility and distinct chemical surroundings (*Van Zundert et al., 2016*).

The performance of HADDOCK protocol depends on the number of models generated at each step. The grading of the clusters is based on the average score of the top four members of each cluster. The score is calculated as:

HADDOCK score = $1.0 \times$ Evdw + $0.2 \times$ Eelec + $1.0 \times$ Edesol + $0.1 \times$ Eair

Where, "Evdw" represents the intermolecular van der Waals energy, "Eelec" is the intermolecular electrostatic energy, where as "Edesol" is an empirical desolvation energy (*Fernández-Recio, Totrov & Abagyan, 2004*), and Eair represents the AIR energy.

Numbering of cluster in the results indicates the magnitude of the cluster. The diverse elements of the HADDOCK score are also described for each cluster on the results web page. The top cluster is the most reliable according to HADDOCK. The more negative results of HADDOCK score and Z score signifies better structures and interaction (HADDOCK 2.4 basic protein–protein docking tutorial, https://www.bonvinlab.org/education/HADDOCK24/HADDOCK24-protein-protein-basic/#analysing-the-results).

For the HADDOCK inputs, the crystal structure of the HLA-C*07:02 (PDB id: 5VGE) HLA-A*30:01 (6J1W), HLA-B*58:01, (5VWH), HLA-B*08:01, (3X13) was retrieved from the RCSB Protein Data Bank (PDB) in the PDB format (*Gras et al., 2010*). PyMOL (Version-2.3.4) was used to remove water and for the retrieval of different chains of HLA allele from the crystal structure, which was in a complex form with protein and a peptide (*PyMOL 2.3.4, 2019*, https://pymol.org/2/). The structure of chain A having the peptide binding cleft was then directly submitted on the HADDOCK 2.4 as protein molecule while PEPFOLD derived structures of predicted epitopes were used as ligands. After registration "easy interface" was selected for docking. In the docking parameter section default parameter was selected. The default parameters can be found on the HADDOCK server website https://wenmr.science.uu.nl/haddock2.4/settings.

Similarly for MHC II epitopes, the crystal structures of HLA-DRB1*01:01 (PDB id:2FSE), HLA-DRB1*01:01 (2FSE) HLA-DRB1*04:01 (5LAX) were retrieved from PDB. PyMOL was used for removing water and the structures of chain A and B were derived to be submitted at HADDOCK as protein molecule. Other procedures similar to MHC I were also followed for docking of MHC II alleles and predicted epitopes.

The 3D structures of the best cluster obtained from the HADDOCK results were visualized using PyMOL (Version-2.3.4). For 2D interaction studies the Discovery studio visualizer (Version: v20.1.0.19295) was used for the MHC I epitopes (*Dassault Systèmes, 2020*), on the other hand LigPlot + (Version: Ligplot+ v.1.4.5) was used for the MHC II epitopes (*Wallace, Laskowski & Thornton, 1995*). LigPlot was used for MHC II epitopes as the discovery studio visualize has a limit of 1000 atoms for epitope.

## Re-docking and validation of the docking methods

For validating docking methodologies the crystal structure of HLA molecules and the corresponding epitope as available in the PDB were selected for re-docking. The crystal structures of the following PDB IDs (i) 5VGE (ii) 6J1W (iii) 5VWH (iv) 3X13, (v) 3C9N (vi) 2FSE and (vii) 5LAX were taken. Then the structures of the HLA molecules and the corresponding peptide were retrieved by using PYMOL. The chain "A" for MHC I

allele and chains "A" and "B" from MHC II allele were submitted as protein molecules as done for the predicted epitope dockings above. In the re-docking, however, the peptides derived from the above crystal structures were used as ligands. In the next steps all the above procedures used for the predicted MHC I and MHC II epitope docking (HADDOCK 2.4 protocol) were followed for re-docking and the best cluster structures were visualized using PYMOL and Discovery Studio/ LIGPLOT+.

### Identification of the B cell epitope

The optimum B-cell epitope identification is the crucial step for epitope-based vaccine design. The B-cell epitopes were identified from the SARS-CoV-2 proteins utilizing the web based server BepiPred-2.0 (http://www.cbs.dtu.dk/services/BepiPred/) (*Jespersen et al., 2017*) and LBtope methods (http://crdd.osdd.net/raghava//lbtope/) (*Singh, Ansari & Raghava, 2013*). BepiPred-2.0 might be viewed as the prime and up-to-date B-cell epitope prediction strategy as it exhibit remarkable solution on both epitope data obtained from a vast number of linear epitopes taken from the IEDB database and on structural data of epitope derived from crystallography studies. LBtope is other robust tool for linear B-cell epitope prediction. It was developed on the basis of experimentally proven non B-cell epitopes derived from IEDB database.

The ElliPro (http://tools.iedb.org/ellipro/) tool was used for the prediction of conformational or discontinuous B-cell epitopes (*Ponomarenko et al., 2008*). ElliPro is considered as most comprehensive method that can identify both the conformational and linear epitopes on the basis of 3-dimensional structure and provides the result score as a protrusion index (PI) (*Ponomarenko et al., 2008*). The specifications for conformational epitope prediction were fixed at 0.8 for minimum score and seven Angstrom (Å) for maximum distance.

### Assessment of physicochemical properties

ExPASy ProtParam tools (https://web.expasy.org/protparam/) was used for the assessment of various physiochemical properties of SARS-CoV-2 proteins and the potential vaccine candidates. Properties like amino acid composition, molecular weight, extinction coefficient, isoelectric point (pI), instability index, aliphatic index, stability (in bacterial, yeast, and mammalian system) grand average hydropathicity (GRAVY) value was identified by ExPASy ProtParam (*Gasteiger et al., 2005*).

## RESULTS

### Retrieval of SARS-CoV-2 proteins and determination of antigenicity of structural proteins

A total of ten protein sequences of SARS-CoV-2 (orf1ab polyprotein, surface glycoprotein or spike protein, orf3a protein, envelope protein, membrane glycoprotein, orf6, orf7a, orf8 proteins and nucleocapsid phosphoprotein) was retrieved from viPR database, out of which six confirmed structural protein (surface glycoprotein, orf3a protein, envelope protein, membrane glycoprotein, orf6 protein, nucleocapsid phosphoprotein) was selected for the epitope-based vaccine designing. Antigenicity analysis of all the six structural
proteins was performed by Vaxijen server. The Vaxijen score of all these proteins were above threshold level, ≥0.4 (Table S1), thus all six selected proteins were antigenic in nature. Highest Vaxijen score was observed for Orf 6 protein (0.6131) and minimum was found in case of surface glycoprotein (0.4646).

## T-cell epitope prediction

The web based server NetCTL 1.2 was used for the identification of CD8[+] T-cell epitopes and the combined score was considered for the selection of epitopes. From the protein sequences of S, ORF3a, E, M, ORF6, and N proteins, the server predicted a total of 83, 33, 10, 31, 4 and 26 epitopes, respectively (Table S2).

## Analysis of antigenicity and immunogenicity

All the 187 identified T-cell epitopes were then evaluated for antigenicity by the VaxiJen server and then for immunogenicity by IEDB server. However, only 97 epitopes from all the six structural proteins were regarded as antigenic based on VaxiJen scores. Similarly, 106 epitopes had immunogenicity values more than the threshold value when analyzed by IEDB tool (Table S2). Altogether 82 epitopes were selected on the basis of positive scores for both antigenicity and immunogenicity (Table S3). The HLA-binding affinity of the epitopes is described by the IC50 nM unit. Higher binding affinity of the epitopes with the MHC class I molecule is reflected by the lower IC50 value. Therefore, IC50 values less than 250nM (IC50 < 250) were fixed for securing higher binding affinity. The IC50 values of the epitopes were determined by the IEDB tool (Table S3). These selected epitopes were then subjected to evaluation of conservancies. We eventually selected 38 epitopes from all the six structural proteins that had a conservancy scores greater than 65 % (Table 1).

## Evaluation of allergenicity and toxicity

The allergenicity determination of the potential epitopes is a critical step in vaccine design. Therefore, Allergen FP 1.0 server and AllerTOP v. 2.0 were used for identifying the allergens in the T cell epitopes. About one third of the epitopes were non-allergenic, while remaining two third were allergic, when the tool, Allergen FP 1.0 was used for evaluation. However, when AllerTOP v. 2.0 was used for the identification of allergenicity, only ten epitopes were found to be allergenic in nature. The server recognized thirteen epitopes from the proteins as non-allergen (Table 1). All the predicted epitopes of MHC-1 from structural proteins of SARS-CoV-2 were indicated as non-toxic, when ToxinPred was used for the toxicity assessment (Table 1).

## Selection of potential MHC-I epitopes for vaccine design

The potential 38 CD8[+] T cell epitopes from six structural proteins were finally evaluated for all the above parameters simultaneously for determination of most suitable vaccine candidates.

Among the S protein epitopes, "YQPYRVVVL" exhibited high binding affinity for seven MHC-I molecules viz; (i) HLA-C*12:03 (37.77) (ii) HLA-A*02:06 (68.16), (iii) HLA-B*39:01 (75.92), (iv) HLA-B*15:02 (92.94), (v) HLA-B*15:01 (181.97), (vi) HLA-C*14:02 (198.89), (vii) HLA-C*03:03 (199.8) (Table S3). It had the VaxiJen score of 0.5964 and

**Table 1 Potential T cell epitopes from different structural proteins from SARS-CoV-2.** The potential T-cell epitopes with interacting MHC-I alleles and antigenicity, immunogenicity and conservancy scores derived from structural proteins of SARS-CoV-2. The most promising proposed vaccine epitopes are highlighted.

| Epitopes | Position | Antigenicity (Vaxijen Score) | MHC-I alleles | Immunogenicity | Conservancy (%) | Allergenicity | | Toxicity |
|---|---|---|---|---|---|---|---|---|
| | | | | | | AllerTOP | AllergenFP | |
| Surface glycoprotein (S) | | | | | | | | |
| YQPYRVVVL | 505–513 | 0.5964 | HLA-C*12:03,HLA-A*02:06, HLA-B*39:01,HLA-B*15:02, HLA-B*15:01,HLA-C*14:02, HLA-C*03:03 | 0.14090 | 100.00 | NO | NO | NT |
| PYRVVVLSF | 507–515 | 1.0281 | HLA-C*14:02,HLA-A*23:01, HLA-C*12:03,HLA-C*07:02, HLA-B*15:02,HLA-A*24:02 | 0.03138 | 100.00 | NO | NO | NT |
| AEIRASANL | 1,016–1,024 | 0.7082 | HLA-C*03:03,HLA-B*40:01, HLA-B*15:02,HLA-C*12:03, HLA-B*40:02,HLA-B*44:03 | 0.00689 | 100.00 | NO | YES | NT |
| FLHVTYVPA | 1,062–1,070 | 1.3346 | HLA-C*03:03,HLA-C*14:02, HLA-C*12:03,HLA-A*02:01, HLA-A*02:06,HLA-B*15:02 | 0.11472 | 88.89 | YES | YES | NT |
| IAIPTNFTI | 712–720 | 0.7052 | HLA-C*03:03,HLA-C*12:03, HLA-B*58:01,HLA-B*53:01, HLA-C*15:02,HLA-A*02:06 | 0.18523 | 88.89 | NO | YES | NT |
| WPWYIWLGF | 1,212–1,220 | 1.4953 | HLA-B*35:01,,HLA-B*53:01, HLA-C*12:03,HLA-B*07:02, HLA-B*15:02 | 0.41673 | 88.89 | YES | YES | NT |
| QYIKWPWYI | 1,208–1,216 | 1.4177 | HLA-A*23:01,HLA-C*12:03, HLA-C*14:02,HLA-A*24:02, HLA-C*03:03 | 0.21624 | 88.89 | YES | NO | NT |
| GQTGKIADY | 413–421 | 1.4019 | HLA-C*03:03,HLA-C*12:03, HLA-A*30:02 | 0.00796 | 88.89 | NO | YES | NT |
| GVYFASTEK | 89–97 | 0.7112 | HLA-C*03:03,HLA-A*11:01, HLA-C*12:03,HLA-A*03:01, HLA-C*14:02,HLA-C*15:02, HLA-A*68:01,HLA-A*30:01 | 0.09023 | 77.78 | NO | NO | NT |
| VTYVPAQEK | 1,065–1,073 | 0.8132 | HLA-C*03:03,HLA-C*15:02, HLA-C*12:03,HLA-C*14:02, HLA-A*11:01,HLA-A*03:01, HLA-A*30:01 | 0.02711 | 77.78 | YES | YES | NT |
| PFFSNVTWF | 57–65 | 0.6638 | HLA-C*12:03,HLA-C*14:02, HLA-C*07:02,HLA-B*15:02, HLA-A*23:01,HLA-C*03:03 | 0.06627 | 77.78 | YES | YES | NT |
| QLTPTWRVY | 628–636 | 1.2119 | HLA-C*03:03,HLA-C*12:03, HLA-C*14:02,HLA-B*15:02, HLA-C*07:02 | 0.31555 | 77.78 | NO | NO | NT |
| VYAWNRKRI | 350–358 | 0.5003 | HLA-C*14:02,HLA-C*12:03, HLA-C*03:03,HLA-A*23:01, HLA-A*24:02 | 0.12625 | 77.78 | YES | YES | NT |
| Orf3A Protein | | | | | | | | |
| LKKRWQLAL | 65–73 | 1.0692 | HLA-C*12:03,HLA-B*15:02, HLA-C*03:03 | 0.10224 | 88.89 | NO | NO | NT |

| Epitopes | Position | Antigenicity (Vaxijen Score) | MHC-I alleles | Immunogenicity | Conservancy (%) | Allergenicity | | Toxicity |
|---|---|---|---|---|---|---|---|---|
| | | | | | | AllerTOP | AllergenFP | |
| HVTFFIYNK | 227–235 | 0.9862 | HLA-A*68:01, HLA-A*11:01, HLA-C*12:03, HLA-A*30:01, HLA-A*31:01, HLA-C*03:03, HLA-A*03:01 | 0.36278 | 66.67 | NO | NO | NT |
| YQIGGYTEK | 184–192 | 1.0504 | HLA-C*12:03,HLA-A*02:06, HLA-C*03:03 | 0.19808 | 77.78 | NO | YES | NT |
| Envelope protein (E) | | | | | | | | |
| LLFLAFVVF | 18–26 | 0.8144 | HLA-B*15:01,HLA-A*32:01, HLA-C*12:03,HLA-C*14:02, HLA-B*15:02,HLA-C*03:03, HLA-A*02:06 | 0.23410 | 100 | NO | YES | NT |
| FLLVTLAIL | 26–34 | 0.9645 | HLA-C*03:03,HLA-A*02:01, HLA-B*15:02,HLA-C*14:02, HLA-A*02:06,HLA-C*12:03 | 0.17608 | 100 | NO | YES | NT |
| FLAFVVFLL | 20–28 | 0.5308 | HLA-A*02:01,HLA-A*02:06, HLA-B*15:02,HLA-C*03:03, HLA-A*68:02,HLA-C*12:03 | 0.30188 | 100 | NO | YES | NT |
| VFLLVTLAI | 25–33 | 0.8134 | HLA-C*14:02,HLA-C*12:03, HLA-A*23:01,HLA-C*03:03 | 0.07548 | 100 | NO | YES | NT |
| Membrane glycoprotein (M) | | | | | | | | |
| LAAVYRINW | 67–75 | 1.4322 | HLA-B*58:01,HLA-C*12:03, HLA-C*03:03,HLA-B*53:01, HLA-B*57:01 | 0.20790 | 100.00 | YES | YES | NT |
| LWPVTLACF | 57–65 | 1.1590 | HLA-C*14:02,HLA-C*12:03, HLA-A*24:02,HLA-A*23:01, HLA-B*15:02 | 0.06682 | 100.00 | NO | YES | NT |
| LWLLWPVTL | 54–62 | 0.7197 | HLA-C*03:03,HLA-C*12:03, HLA-B*15:02,HLA-A*23:01, HLA-C*14:02 | 0.24802 | 100.00 | YES | NO | NT |
| FAYANRNRF | 37–45 | 0.7785 | HLA-C*03:03,HLA-C*12:03, HLA-B*15:02,HLA-B*35:01, HLA-C*14:02,HLA-B*53:01 | 0.10537 | 88.89 | YES | YES | NT |
| SYFIASFRL | 94–102 | 0.4821 | HLA-B*15:02,HLA-C*14:02, HLA-C*12:03,HLA-C*07:02, HLA-A*23:01,HLA-A*24:02 | 0.18333 | 88.89 | NO | YES | NT |
| KLIFLWLLW | 50–58 | 0.4968 | HLA-B*58:01,HLA-A*32:01, HLA-C*12:03,HLA-B*57:01 | 0.34287 | 88.89 | NO | YES | NT |
| RFLYIIKLI | 44–52 | 0.4257 | HLA-C*03:03,HLA-C*14:02, HLA-C*12:03,HLA-A*23:01 | 0.05908 | 88.89 | NO | NO | NT |
| LYIIKLIFL | 46–64 | 0.4865 | HLA-C*14:02,HLA-C*03:03, HLA-C*12:03 | 0.13740 | 88.89 | NO | YES | NT |
| LTWICLLQF | 29–37 | 1.1393 | HLA-C*14:02,HLA-B*58:01, HLA-C*12:03,HLA-A*32:01 | 0.06584 | 77.78 | NO | NO | NT |
| ORF6 protein | | | | | | | | |
| HLVDFQVTI | 23–31 | 1.4119 | HLA-C*12:03,HLA-A*29:02, HLA-C*05:01,HLA-A*30:02 | 0.09820 | 100.00 | YES | NO | NT |

(Continued)

| Epitopes | Position | Antigenicity (Vaxijen Score) | MHC-I alleles | Immunogenicity | Conservancy (%) | Allergenicity | | Toxicity |
|---|---|---|---|---|---|---|---|---|
| | | | | | | AllerTOP | AllergenFP | |
| LLIIMRTFK | 3–11 | 0.4377 | HLA-C*12:03,HLA-C*03:03, HLA-A*32:01,HLA-A*02:01 | 0.15600 | 77.78 | NO | YES | NT |
| Nucleocapsid phosphoprotein (N) | | | | | | | | |
| KTFPPTEPK | 361–369 | 0.7571 | HLA-A*30:01,HLA-C*12:03, HLA-C*14:02,HLA-A*03:01, HLA-A*31:01,HLA-A*68:01, HLA-C*03:03,HLA-A*32:01, HLA-A*11:01 | 0.13060 | 100.00 | NO | NO | NT |
| LSPRWYFYY | 104–112 | 1.2832 | HLA-C*12:03,HLA-A*29:02, HLA-A*01:01,HLA-B*15:02 | 0.35734 | 100.00 | NO | YES | NT |
| SPRWYFYYL | 105–113 | 0.7340 | HLA-B*07:02,HLA-B*08:01, HLA-B*15:02,HLA-C*12:03 | 0.34101 | 100.00 | NO | NO | NT |
| QRNAPRITF | 9–17 | 0.4654 | HLA-C*07:02,HLA-C*07:01, HLA-B*15:02,HLA-C*06:02, HLA-C*12:03,HLA-C*03:03, HLA-C*14:02 | 0.21019 | 88.89 | NO | YES | NT |
| DLSPRWYFY | 103–111 | 1.7645 | HLA-C*03:03,HLA-C*12:03, HLA-A*29:02,HLA-C*07:02, HLA-B*15:02,HLA-A*30:02 | 0.25933 | 88.89 | NO | YES | NT |
| TWLTYTGAI | 329–337 | 0.5439 | HLA-C*14:02,HLA-C*03:03, HLA-C*12:03 | 0.11986 | 88.89 | NO | NO | NT |
| SSPDDQIGY | 78–86 | 0.5260 | HLA-C*12:03,HLA-C*07:01, HLA-C*14:02 | 0.06340 | 88.89 | NO | YES | NT |

the immunogenicity score of 0.14090, which were well above the respective threshold values. The conservancy score of the epitope was 100.00% and both the allergenicity prediction tools identified this epitope as non allergenic. Furthermore, it was nontoxic as determined by the toxicity analysis tool used in this study. Similarly, three more epitopes from spike protein viz; "QLTPTWRVY", "PYRVVVLSF" and "GVYFASTEK" exhibited desired values for the above parameters (highlighted with yellow color in Table 1).

Next, the epitopes from ORF3a was selected on the basis of outcomes of the parameters evaluated in the study. The epitope "HVTFFIYNK" also showed binding capabilities with seven MHC class I alleles. The VaxiJen score of 0.9862 and immunogenicity score of 0.36278 was found for the epitope "HVTFFIYNK". Further, with the conservancy value of 66.67% and being non-allergenic and nontoxic this epitope can be regarded as best vaccine candidate from ORF3a protein. Likewise one more epitope "LKKRWQLAL" was marked as potential vaccine candidates based on their scores gathered during the analysis by the computational tools (Table 1; Table S3).

In contrast to epitopes from spike and ORF3a proteins, although E protein epitope "LLFLAFVVF" exhibited binding affinity with seven MHC I alleles and high antigenicity, immunogenicity, it could not pass the allergenicity evaluation. Thus, no epitope from envelope protein could be regarded as potential vaccine candidates.

Among epitopes from M protein, "LTWICLLQF" had high antigenicity score of 1.1393 and immunogenicity score of 0.06584. "LTWICLLQF" exhibited binding to (i) HLA-C*14:02 (129.6), (ii) HLA-B*58:01, (141.02), (iii) HLA-C*12:03 (166.42), (iv) HLA-A*32:01 (245.43) molecules and had a conservancy score of 77.78%. Furthermore, the epitope was non-allergenic and had no toxicity, thus can be regarded as one of the best vaccine candidates from M protein. One more epitope "RFLYIIKLI" had better scores in the computational analysis performed for the evaluation of vaccine potential (Table 1; Table S3).

Similar to envelope protein, ORF6 epitope also did not show any promising vaccine candidate that could fulfill all the criteria evaluated in our study (Table 1).

Last of the selected structural protein, nucleocapsid protein showed three promising epitopes, when evaluated by the computational tools. One of three epitopes; "KTFPPTEPK" displayed significant binding affinities with nine MHC-1 molecules. It had VaxiJen score of 0.7571 and immunogenicity score of 0.13060. The conservancy was 100.00% for this epitope and was categorized as non-allergen and non-toxic by the computational tools. Two other epitopes "SPRWYFYYL" and "TWLTYTGAI" also fulfilled all the criteria analyzed in the study for the determination of vaccine potential (Table 1; Table S3).

## Analysis of population coverage

The distribution of MHC HLA alleles varies across various geographic territories and ethnic classes throughout the world. Consequently, consideration of population coverage is essential prerequisite for designing an effective vaccine. IEDB population coverage tool was thus used to predict the population coverage of all the shortlisted T-cell epitopes (Table 1) and their respective MHC-I-binding alleles. Remarkable population coverage was identified for the epitopes in different geographic regions of the world (Fig. S1; Table S4).

## Prediction of MHC II epitopes

The MHC II epitopes of 15-m length were derived from the sequences of CD8[+] T cell epitopes and were evaluated on the basis of IC50 scores. The promising CD8[+] T cell epitopes "YQPYRVVVL" and "QLTPTWRVY" were evaluated first. The analysis revealed that the epitope sequence "YQPYRVVVL" was present as the core sequence in more than fifty predicted MHC II epitopes, whereas the epitope "QLTPTWRVY" was found as the core sequence of a single CD4+ T-cell epitope (Table S5). The MHC II epitopes containing the core peptide "LKKRWQLAL" from ORF3a was found to be present as the core sequence in 44 predicted MHC II epitopes (Table S5).

More than forty CD4[+] T cell epitopes having the core sequence "LTWICLLQF" derived from M protein had a range of binding affinities with IC50 values between 61 and 2,801 nM (Table S5).

As the computational analysis of ORF6 and envelope protein did not result in any potential CD8[+] T cell epitope, the MHC II epitopes derived from these proteins were not considered further.

Lastly, the search for MHC II binding epitopes using the core peptide "KTFPPTEPK" could not result in the potential epitopes in the acceptable range of the IC50 value 1–3,000 nM.

## Docking simulation analysis

The CD4$^+$ T cell epitopes which were considered to be potential vaccine candidates based on appropriate values obtained during the analysis by the computational tools were used in the docking simulation studies. The binding models of epitopes and their respective HLA molecules (both class I and class II) were generated by taking advantage of HADDOCK 2.4. The tool generated clusters and the numbering of cluster reflected the size of the cluster. The various components of the HADDOCK results like HADDOCK score, Cluster size, Root mean square deviation (RMSD) from the overall lowest-energy structure, Van der Waals energy, Electrostatic energy, Desolvation energy, Restraints violation energy, Buried Surface Area, and Z-Score were reported for each cluster on the results web page. Irrespective of the number of cluster, the top cluster is considered as most reliable according to HADDOCK. Therefore, the first cluster from the result displayed by HADDOCK server was selected for visualization of structures. The more negative results of HADDOCK score and Z score signifies better structures and interaction. At the first instance two promising MHC I epitopes viz; "PYRVVVLSF", "QLTPTWRVY" from surface glycoprotein was used for the docking with HLA-C*07:02. The HADDOCK score of the first cluster was −30.4 ± 7.5 and the Z score was −1.2 indicating the proper docking solution for HLA-C*07:02 and epitope PYRVVVLSF (Table 2). Similarly the second epitope "QLTPTWRVY" from S protein also had negative values for both the HADDOCK score and Z score indicating the cluster as a good docking solution. In the next HADDOCK analysis epitope "HVTFFIYNK" from ORF3a was used along with the HLA-A*30:01. The best cluster had the HADDOCK score of −65.5 ± 7.7 and the Z score of −1.8, suggesting proper docking results. Another round of docking studies was performed with the MHC I peptide "LTWICLLQF" obtained from the membrane protein and the structure of HLA-B*58:01 molecule derived from PDB. HADDOCK and Z score of the best cluster of this pair were also promising and could be used for the structure visualization. Finally epitope "SPRWYFYYL" from nucleocapsid protein was selected for docking with the HLA-B*08:01 molecule, which resulted in the HADDOCK score of −29.3 ± 3.2 and the Z score of −2.1 (Table 2). In the next step model structures of all the above best clusters obtained in the HADDOCK results were downloaded. Then 3D model of the clusters were visualized by PYMOL and Discovery Studio was utilized for getting the 2D interaction map (Fig. 2). All the 3D and 2D interaction map indicated binding in the antigen binding groove thus providing proper docking solutions by HADDOCK.

Altogether three promising MHC II epitopes were used for HADDOCK docking analysis. First docking analysis was performed using HLA-DRB1*01:01 structure obtained from PDB and the structure of MHC class II epitope "TNGVGYQPYRVVVLS" (from S protein) predicted using PEPFOLD tool. The HADDOCK protocol produced the best cluster with the HADDOCK score of −38.1 ± 9.9 and the Z score of −2.2, which indicated optimum solution of docking. The second docking analysis between the MHC-II allele

**Table 2 Results of docking studies performed using HADDOCK 2.4 with selected T cell epitopes and corresponding HLA molecules.**

| | MHC-1 | | | | | MHC-2 | | |
|---|---|---|---|---|---|---|---|---|
| **PDB id** | **5VGE** | **5VGE** | **6J1W** | **5VWH** | **3X13** | **2FSE** | **2FSE** | **5LAX** |
| **HLA molecule** | HLA-C*07:02 | HLA-C*07:02 | HLA-A*30:01 | HLA-B*58:01 | HLA-B*08:01 | HLA-DRB1*01:01 | HLA-DRB1*01:01 | HLA-DRB1*04:01 |
| Epitope | PYRVVVLSF | QLTPTWRVY | HVTFFIYNK | LTWICLLQF | SPRWYFYYL | TNGVGYQPYR VVVLS | ITLKKRWQLAL SKGV | LFLTWICLLQ FAYAN |
| HADDOCK score | −30.4 ± 7.5 | −8.5 ± 5.3 | −65.5 ± 7.7 | −10.7 ± 1.0 | −29.3 ± 3.2 | −38.1 ± 9.9 | −23.5 ± 8.5 | −78.4 ± 10.7 |
| Cluster size | 13 | 38 | 44 | 105 | 84 | 8 | 4 | 57 |
| RMSD from the overall lowest-energy structure | 2.4 ± 0.1 | 0.5 ± 0.3 | 0.5 ± 0.3 | 1.5 ± 0.2 | 0.6 ± 0.4 | 0.8 ± 0.5 | 0.4 ± 0.3 | 0.5 ± 0.3 |
| Van der Waals energy | −46.6 ± 4.5 | −58.8 ± 8.1 | −64.5 ± 6.3 | −52.2 ± 3.4 | −62.4 ± 4.9 | −65.7 ± 4.1 | −52.2 ± 4.6 | −74.9 ± 7.2 |
| Electrostatic energy | −199.1 ± 10.1 | −72.7 ± 27.1 | −269.7 ± 30.4 | −41.7 ± 9.2 | −158.1 ± 11.5 | −81.1 ± 24.4 | −157.7 ± 51.4 | −48.0 ± 6.7 |
| Desolvation energy | −29.7 ± 0.8 | −20.8 ± 1.3 | −44.8 ± 3.1 | −37.6 ± 2.7 | −45.8 ± 3.3 | −19.3 ± 1.1 | −14.9 ± 2.2 | −60.4 ± 2.1 |
| Restraints violation energy | 856.7 ± 73.0 | 856.5 ± 33.6 | 977.1 ± 59.8 | 875.0 ± 34.7 | 1105.8 ± 27.7 | 632.2 ± 117.2 | 752.0 ± 91.9 | 664.6 ± 44.5 |
| Buried Surface Area | 1803.4 ± 49.8 | 1575.6 ± 86.2 | 1840.6 ± 27.6 | 1471.4 ± 47.3 | 1612.3 ± 51.2 | 1681.7 ± 49.6 | 1722.6 ± 47.8 | 1985.6 ± 17.1 |
| Z-Score | −1.2 | −1.3 | −1.8 | −1.4 | −2.1 | −2.2 | −1.4 | −1.6 |

(HLA-DRB1*01:01) and epitope "ITLKKRWQLALSKGV" from ORF3a protein also resulted in desired negative values of HADDOCK and Z scores The last HADDOCK docking examination was performed with the MHC II allele (HLA-DRB1*04:01) and the epitope "LFLTWICLLQFAYAN" from membrane glycoprotein of SARS-CoV-2, which revealed a HADDOCK score of −78.4 ± 10.7 and the *Z* score of −1.6 (Table 2). Similar to MHC I, the model structures of all the above three best clusters obtained in the HADDOCK results were downloaded from the HADDOCK result page. 3D models of the clusters were visualized by PYMOL and Discovery Studio was implied for getting the 2D interaction maps of all three docking solutions (Fig. 3). All three 3D and 2D interaction images exhibited proper MHC II allele and epitope binding suggesting appropriate docking solutions by HADDOCK (Fig. 3).

### Re-docking and validation of docking methods

For the validation of docking methodologies crystal structures of seven HLA alleles and the corresponding epitopes as described in method section were selected and all the procedures described for HADDOCK 2.4 were followed for re-docking. The HADDOCK

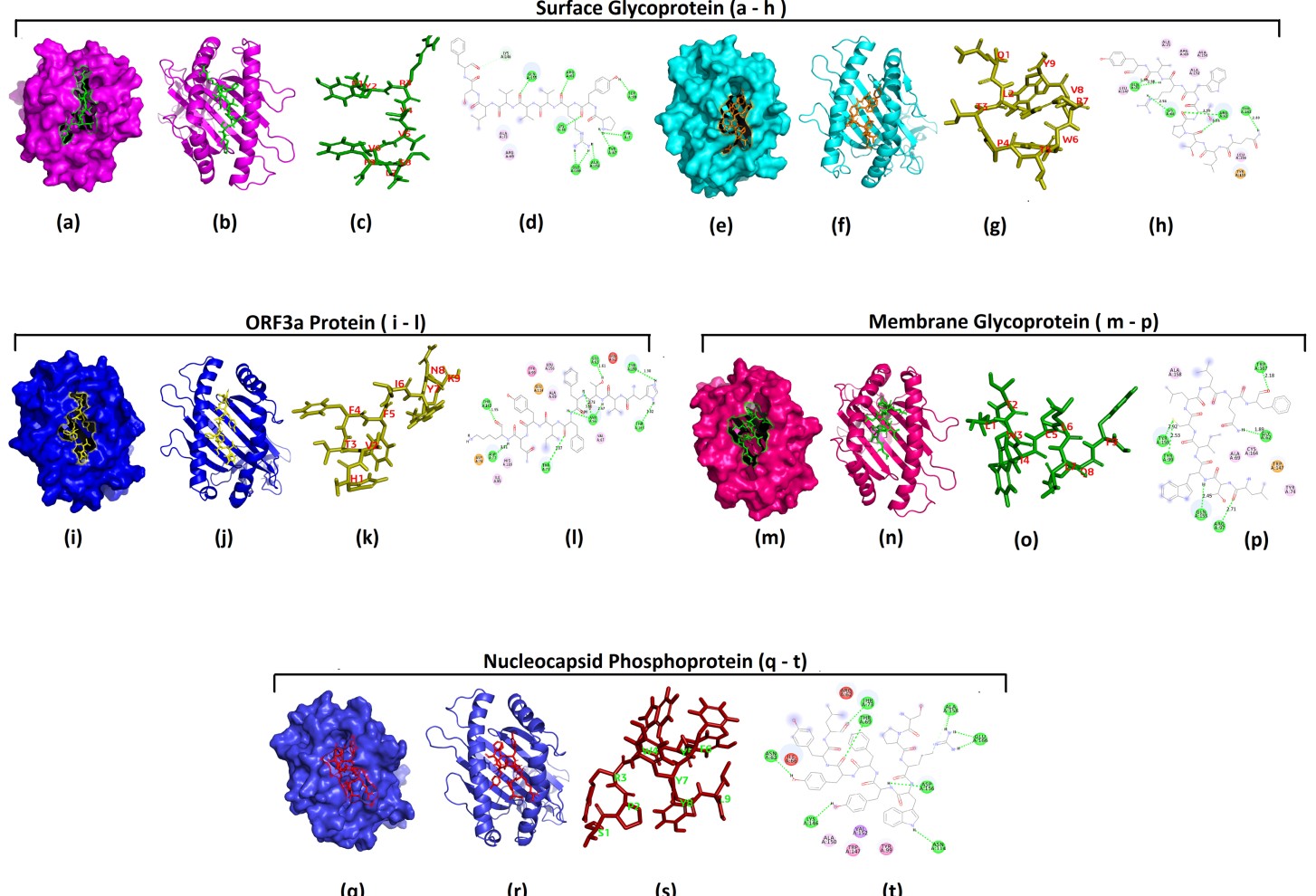

**Figure 2 Docking simulation of CD8+ T cell epitopes with MHCI alleles.** Docking simulation study of MHC I epitopes: HADDOCK 2.4 was used for the docking of MHC I epitopes with the corresponding HLA allele. 3D structures of best clusters were then visualized using PYMOL 2.3.4 and 2D interaction map was visualized using Discovery Studio tools. Images pertaining to epitopes from Surface Glycoprotein or spike protein (A–H); (A) 3D structure of surface of the chain "A" of MHC-I HLA allele "HLA-C*07:02" and the sticky form of epitope "PYRVVVLSF" from spike protein; (B) epitope "PYRVVVLSF" and the chain "A" of HLA-C*07:02 in 3D cartoon structure; (C) sticky form of epitope "PYRVVVLSF" with positions of residues; (D) **2D interaction map of epitope "PYRVVVLSF" and the residues from chain "A" of HLA-C*07:02; (E) 3D structure of surface of the chain "A" "HLA-C*07:02" and the sticky form of epitope "QLTPTWRVY" from spike protein (F) epitope "QLTPTWRVY" and the chain "A" of "HLA-C*07:02" in 3D cartoon structure; (G) sticky form of epitope "QLTPTWRVY" with residue's position. (H) 2D interaction map of epitope "QLTPTWRVY" and the residues from chain "A" of HLA-C*07:02. Images pertaining to an epitope from ORF3a protein (I–L); (I) 3D structure of surface of the chain "A" of "HLA-A*30:01" and the sticky form of epitope "HVTFFIYNK" from ORF3a protein; (J) epitope "HVTFFIYNK" and the chain "A" of "HLA-A*30:01" in 3D cartoon structure; (K) Sticky form of epitope "HVTFFIYNK" with positions of residues; (L) 2D interaction map of epitope "HVTFFIYNK" and the residues from chain "A" of HLA-A*30:01. Images of an epitope from Membrane glycoprotein (M–P); (M) 3D structure of surface of the chain "A" of "HLA-B*58:01" and the sticky form of epitope "LTWICLLQF" from membrane protein; (N) Epitope "LTWICLLQF" and the chain "A" of "HLA-B*58:01" in 3D cartoon structure; (O) sticky form of epitope "LTWICLLQF" with residue's position; (P) 2D interaction map of epitope "LTWICLLQF" and the residues from chain "A" of HLA-B*58:01. Images of an epitope from Nucleocapsid phosphoprotein (Q–T); (Q) 3D structure of surface of the chain "A" of "HLA-B*08:01" and the sticky form of epitope "SPRWYFYYL" from Nucleocapsid phosphoprotein; (R) Epitope "SPRWYFYYL" and the chain "A" of "HLA-B*08:01" in the 3D cartoon structure; (S) sticky form of epitope "SPRWYFYYL" with residue's position; (T) 2D interaction map of epitope "SPRWYFYYL" and the residues from chain "A" of HLA-B*8:01. **In all 2D interaction diagrams, colors depict different types of interactions: (i) green color—hydrogen bond (classical, Non classical); (ii) orange color—electrostatic (salt Bridge, Charge, pi-charge); (iii) pink color—hydrophobic (Pi hydrophobic, Alkyl hydrophobic, Mixed pi/Alkyl hydrophobic); (iv) white color—carbon hydrogen bond; (v) red color—unfavorable (Charge Replusion, Acceptor/Donor clash).

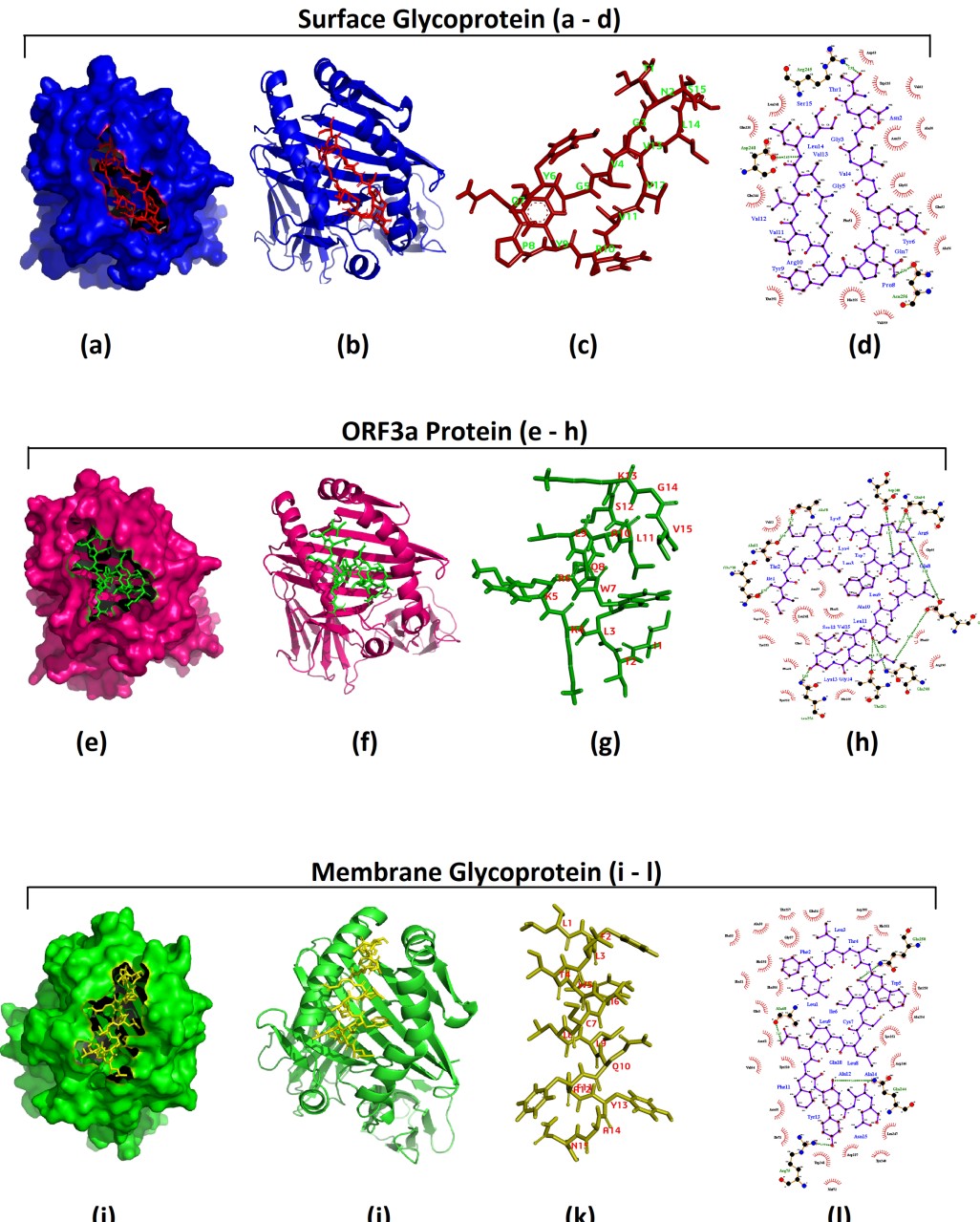

**Figure 3 Docking simulation of CD4+ T cell epitopes with MHC II alleles.** Docking simulation study of MHC II epitopes: HADDOCK 2.4 was used for the docking of MHC II epitopes with the corresponding HLA allele. 3D structures of best clusters were then visualized using PYMOL 2.3.4 and 2D interaction map was visualized using Ligplot⁺ v.1.4.5. Images pertaining to epitopes from Surface Glycoprotein or spike protein (A–D); (A) 3D structure of surface of the chain "A" and "B" from MHC-II HLA allele, "HLA-DRB1*01:01" and the sticky form of epitope "TNGVGYQPYRVVVLS" from spike protein; (B) epitope "TNGVGYQPYRVVVLS" and both the chains from HLA-DRB1*01:01 in 3D cartoon structure; (C) sticky form of epitope "TNGVGYQPYRVVVLS" with positions of residues; (D) **2D interaction map of epitope "TNGVGYQPYRVVVLS" and the residues from both chain "A" and "B" of HLA-DRB1*01:01. Images pertaining to epitopes from ORF3a protein (E–H): (E) 3D structure of surface of the two chains A and B of "HLA-DRB1*01:01" and the sticky form of epitope "ITLKKRWQ-LALSKGV" from ORF3a protein; (F) epitope "ITLKKRWQLALSKGV" and both the chains of HLA-DRB1*01:01 in 3D cartoon structure; (G) sticky form of epitope "ITLKKRWQLALSKGV" with positions

**Figure 3** (continued)
of residues; (H) 2D interaction map of epitope "ITLKKRWQLALSKGV" and the residues from the
chains A and B of HLA-DRB1*01:01. Images of an epitope from Membrane glycoprotein (I–L); (I) 3D
structure of surface of the chains A and B from "HLA-DRB1*04:01" allele and the sticky form of epitope
"LFLTWICLLQFAYAN" from membrane glycoprotein; (J) epitope "LFLTWICLLQFAYAN" and the
chains A and B from HLA-DRB1*04:01 in 3D cartoon structure; (K) sticky form of epitope
"LFLTWICLLQFAYAN" with positions of residues; (L) 2D interaction map of epitope
"LFLTWICLLQFAYAN" and the residues from A to B chain of HLA-DRB1*04:01. **In all 2D inter-
action diagram colors depict different types of bonds: (i) purple—ligand bonds; (ii) orange—non-ligand
bonds; (iii) olive green—hydrogen bonds; (iv) brick red—hydrophobic bonds. The atoms are also
depicted by color in; blue—nitrogen; red—oxygen; black—carbon; yellow—sulphur; turquoise—water;
purple—phosphorous; pink—metal; lime green—other atom. Brick red denotes hydrophobic residue.

scores were in the range of −29.6 ± 7.8 to −65.0 ± 2.1 fo the MHC I epitopes while it
was −92.1 ± 5.8 and −96.7 ± 5.6 for the two MHC II epitopes. The $Z$ scores were also in the
range of −1.0 to −2.3 for MHC I structures while it was 0.0 and −1.5 for MHC II
alleles (Table S6). Then 3D and 2D structures of the best clusters were visualized using
PYMOL and Discovery studio/ LigPLOT+. The results indicated proper docking
solutions provided by HADDOCK and the structures were similar as available in the PDB
(Fig. S2). The findings of the re-docking with the known epitopes and HLA alleles
validated the docking methodology adopted in the present study

## Analysis of linear and conformational B-cell epitopes

B-cell epitope is a segment of an antigen recognized in a humoral immune response by
either a specific B-cell receptor or by the evoked antibody (Peters et al., 2005; Sun et al.,
2013). The B-cell epitopes are categorized into two distinct groups as (i) continuous or
linear and (ii) discontinuous or conformational B-cell epitopes. One of the significant
steps of epitope-based vaccine design is the identification of B-cell epitopes from the
antigenic proteins of pathogens. Consequently, the web server based computational tools,
BepiPred-2.0 and LBtope were used to find out B-cell vaccine candidates in the different
proteins of SARS-CoV-2.

The BepiPred-2.0 generated fair number of linear B-cell epitopes from the S protein of
SARS-CoV-2. Among these linear epitopes eleven were non-antigenic as predicted by the
VaxiJen v2.0 server and had conservancy level between 56.25% and 97.50% (Table S7).
Accordingly, these epitopes could not be considered as prospective vaccine candidates.
Conversely, six epitopes; (i) GQSKRVDFC, (ii) VEAEVQI, (iii) SCCKFDEDDSEPVLKGVKL,
(iv) GDEVRQIAPGQTGKIADYNYK, (v) YQTSNFRVQP and (vi) NSASFSTFKCY
GVSPTKLND LCFTNV can be regarded as vaccine candidates due to their antigenicity and
high conservancy scores (Table 3). Nevertheless, the epitope "SCCKFDEDDSEPVLKGVKL"
being toxic in nature could not be considered as the potential vaccine candidate. Based
on the results of allergenicity (AllerTOP 2.0 and AllergenFP v. 1.0) and toxicity, epitope
"NSASFSTFKCYGVSPTKLNDLCFTNV" could be considered as best potential linear B-cell
epitope for vaccine design (highlighted in Table 3). Over twenty linear B-cell epitopes were
recognized from the S protein using LBtope (Table S7). Altogether seven epitopes were
non-antigenic and cannot be considered as good vaccine candidates. The epitope

**Table 3 Linear B-cell epitopes of SARS-CoV-2.** (A) Linear B-cell epitopes from structural proteins of SARS-CoV-2 predicted by BepiPred-2.0. (B) Linear B-cell epitopes from structural proteins of SARS-CoV-2 predicted by LBtope. The highlighted epitopes are the best epitopes and promising vaccine candidates.

| B-cell epitopes | Position | Antigenicity score | Conservancy (%) | Toxicity | Allergenicity | |
|---|---|---|---|---|---|---|
| | | | | | AllerTOP 2.0 | AllergenFP 1.0 |
| (A) Linear B-cell epitopes from structural proteins of SARS-CoV-2 predicted by BepiPred-2.0 | | | | | | |
| Surface glycoprotein | | | | | | |
| GQSKRVDFC | 1,035–1,043 | 1.779 | 100.00 | NT | YES | YES |
| NSASFSTFKCYGVSPTKLNDLCFTNV | 370–395 | 1.3609 | 84.62 | NT | NO | NO |
| GDEVRQIAPGQTGKIADYNYK | 404–424 | 1.3212 | 90.48 | NT | YES | YES |
| YQTSNFRVQP | 313–322 | 1.1866 | 90.00 | NT | NO | YES |
| VNCTEVP | 615–621 | 1.129 | 71.43 | NT | YES | YES |
| NNLDSKVGGNYNY | 439–451 | 0.9437 | 53.85 | NT | YES | NO |
| DLEGKQGNFKNLRE | 178–191 | 0.9256 | 64.29 | NT | YES | NO |
| VEAEVQI | 987–993 | 0.8205 | 100.00 | NT | YES | YES |
| QCVNLTTRTQLPPAYTNSFTRGV | 14–36 | 0.7515 | 26.09 | NT | YES | NO |
| FSNVTWFHAIHVSGTNGTKRFDN | 59–81 | 0.6767 | 39.13 | NT | YES | YES |
| YLTPGDSSSGWTA | 248–260 | 0.627 | 38.46 | NT | NO | NO |
| SCCKFDEDDSEPVLKGVKL | 1,252–1,270 | 0.6085 | 100.00 | TOXIN | YES | NO |
| AYTMSLGAENSVAYSN | 694–709 | 0.6003 | 81.25 | NT | YES | NO |
| VEGFNCYFPLQ | 483–493 | 0.5612 | 45.45 | NT | YES | YES |
| VNNSYECDIP | 656–665 | 0.5327 | 80.00 | NT | NO | YES |
| LGVYYHKNNKSWMESEFRVYSSA | 141–163 | 0.4829 | 21.74 | NT | NO | YES |
| FYEPQIITTD | 1,109–1,118 | 0.4179 | 80.00 | NT | YES | YES |
| orf3a protein | | | | | | |
| QGEIKDATPSDF | 17–28 | 1.1542 | 33.33 | NT | YES | NO |
| KIITLKKRWQL | 61–71 | 1.0171 | 81.82 | NT | NO | YES |
| Envelope protein | | | | | | |
| YVYSRVKNLNSSRVP | 57–71 | 0.4492 | 80.00 | NT | NO | NO |
| membrane glycoprotein | | | | | | |
| KLGASQRVAGDS | 180–191 | 0.0439 | 83.33 | NT | NO | NO |
| RYRIGNYKLNTDHSSSSDNIA | 198–218 | 0.1635 | 85.71 | NT | NO | YES |
| orf6 protein | | | | | | |
| LTENKYSQLDEEQP | 44–57 | 0.5866 | 57.14 | NT | YES | YES |
| nucleocapsid phosphoprotein | | | | | | |
| HGKEDLKFPRGQGVPINTNSSPDDQ IGYYRRATRRIRGGDGKMKDLS | 59–105 | 0.5773 | 89.36 | NT | NO | NO |
| TLPKGFYAEGSRGGSQASSRSSSRSR NSSRNSTPGSSRGTSPARMAGNGGD | 166–216 | 0.5064 | 88.24 | NT | NO | YES |
| LNQLESKMSGKGQQQQGQTVTKK SAAEASKKPRQKRTATK | 227–266 | 0.5387 | 97.50 | NT | NO | NO |
| RRGPEQTQGNFGDQELIRQGTDYK | 276–299 | 0.6277 | 95.83 | NT | NO | YES |

(Continued)

| B-cell epitopes | Position | Antigenicity score | Conservancy (%) | Toxicity | Allergenicity | |
|---|---|---|---|---|---|---|
| | | | | | AllerTOP 2.0 | AllergenFP 1.0 |

(B) Linear B-cell epitopes from structural proteins of SARSCoV-2 predicted by LBtope. The highlighted epitopes are the best epitopes and promising vaccine candidates

Surface glycoprotein

| B-cell epitopes | Position | Antigenicity score | Conservancy (%) | Toxicity | AllerTOP 2.0 | AllergenFP 1.0 |
|---|---|---|---|---|---|---|
| CYGVSPTKLN | 379–388 | 1.5759 | 90.00 | NT | YES | NO |
| TLEILDITPC | 581–590 | 1.5604 | 80.00 | NT | YES | NO |
| PVLKGVKLHY | 1,263–1,272 | 1.4055 | 100.00 | NT | NO | YES |
| AGAAAYYVGYLQPRT | 260–274 | 0.9134 | 66.67 | NT | NO | NO |
| GFQPTNGVGYQPYRVVVLSF | 496–515 | 0.8857 | 80.00 | NT | YES | NO |
| PFLGVYYHKNNKSW | 139–152 | 0.7487 | 28.57 | NT | NO | NO |
| PINLVRDLPQGFSALEPLVDLPIGI | 209–233 | 0.6961 | 60.00 | NT | YES | NO |
| PLSETKCTLKSFT | 295–307 | 0.6582 | 61.54 | NT | YES | NO |
| RARSVASQ | 683–690 | 0.6389 | 37.50 | NT | NO | NO |
| KVGGNYNYL | 444–452 | 0.5994 | 55.56 | NT | YES | YES |
| VFLVLLPLVSSQCVN | 03–17 | 0.5954 | 33.33 | NT | NO | NO |
| KKSTNLVKNKCV | 528–539 | 0.5949 | 66.67 | TOXIN | YES | NO |
| IQDSLSSTASALGK | 934–947 | 0.5193 | 64.29 | NT | YES | NO |
| SQPFLMDL | 172–179 | 0.4797 | 50.00 | NT | YES | YES |
| orf3a protein | | | | | | |
| EIKDATPSDF | 19–28 | 1.5094 | 40.00 | NT | YES | NO |
| WKCRSKNPLL | 131–140 | 1.2111 | 90.00 | TOXIN | YES | NO |
| Envelope protein | | | | | | |
| YVYSRVKNLNSSRVP | 57–71 | 0.4492 | 72.22 | NO | NO | NO |
| membrane glycoprotein | | | | | | |
| ITVATSRTLSYYKLGASQR | 168–186 | 0.7666 | 100.00 | NT | NO | YES |
| SDNIALL | 214–219 | 0.4677 | 85.71 | NT | NO | YES |
| orf6 protein | | | | | | |
| FHLVDFQVTI | 02–11 | 1.8174 | 100.00 | NT | YES | NO |
| SKSLTENKYSQLDEEQPME | 41–59 | 0.4682 | 57.89 | NT | NO | NO |
| nucleocapsid phosphoprotein | | | | | | |
| DNGPQNQRNAPRITFGGP | 3–20 | 0.4751 | 66.67 | NT | NO | NO |
| GERSGARSKQRRPQGL | 29–45 | 0.5789 | 81.25 | NT | NO | NO |
| DLKFPRGQGVPINTNSSPDDQIGYYRR ATRRIRGGDGKMKDLSPRWYFYYL | 63–113 | 0.6372 | 90.20 | NT | NO | YES |
| DPNFKDQV | 343–350 | 1.7958 | 75.00 | NT | YES | YES |

"AGAAAYYVGYLQPRT" had high antigenicity and high conservancy scores and were not classified as allergen by the tools, hence can be considered as potential vaccine candidate. Compared to the spike protein, only five linear B-cell epitopes from ORF3a protein were identified by BepiPred-2.0 (Table S7). Out of these epitopes, three were non-antigenic as discerned by the VaxiJen v2.0 server and their conservancy scores varied between

57.69% and 74.29%. On the other hand, two epitopes; (i) "QGEIKDATPSDF" and (ii) "KIITLKKRWQL" can be considered as vaccine candidates due to their antigenicity and conservancy score (Table 3). After due consideration of all the factors like allergenicity (AllerTOP 2.0 and AllergenFP v. 1.0) and toxicity, none of the epitope predicted by BepiPred-2.0 could be safely recommended as potential linear B-cell epitopes. Another tool for linear epitope discovery, LBtope, led to the identification of only three linear B-cell epitopes from the ORF3a protein. Although, two epitopes; "EIKDATPSDF" and "WKCRSKNPLL" had fair values for antigenicity and high conservancy score, but owing to its toxicity, it cannot be projected as potential vaccine candidates. The evaluation of antigenicity, conservancy, toxicity, and allergenicity of B-cell epitopes suggested that none of the linear B-cell epitopes from ORF3a could be considered as candidates for peptide-based vaccine design.

When the E protein was investigated using BepiPred-2.0 server, only one epitope; "YVYSRVKNLNSSRVP" was identified as linear B-cell epitope (Table S7). It showed good antigenicity with non-allergenic and non-toxic property and a conservancy score of 80.00%. Similarly, LBtope also showed only one epitope, YVYSRVKNLNSSRVPDLL that too was antigenic, non allergenic and non toxic with conservancy score of 72.22%. Consequently, YVYSRVKNLNSSRVP can be regarded as most potential B-cell epitope candidate from E protein for peptide-based vaccine (Table 3).

The search for potential linear B-cell epitopes from M and ORF6 protein by BepiPred-2.0 and LBtope could not be successful as none of the predicted epitope could satisfy all the criteria evaluated in the present study.

The search of linear B-cell epitopes in N protein by BepiPred-2.0 resulted in identification of eight epitopes (Table S7). Among the predicted epitopes four were reported as non-antigenic by the VaxiJen v2.0 server. Out of the remaining four epitope only two epitopes viz; "HGKEDLKFPRGQGVPINTNSSPDDQIGYYRRATRRIRGGDG KMKDLS", and "LNQLESKMSGKGQQQ QGQTVTKKSAAEASKKPRQKRTATK" could be considered as the potential linear B-cell epitopes for vaccine development. On the other hand only 4 linear B-cell epitopes were predicted by LBtope (Table S7). The analysis of antigenicity, conservancy, toxicity, and allergenicity of B-cell epitopes identified by LBtope revealed that epitopes, DNGPQNQRNAPRITFGGP, GERSGARSKQRRPQGL could be regarded as the most potential B-cell linear epitope (Table 3).

For identifying conformational B-cell epitopes, the ElliPro tool of IEDB was utilized in this study, and a total of eleven discontinuous peptides were identified when the structural proteins of SARS-CoV-2 were used as targets. The ElliPro tool evaluates results based on the protrusion index (PI) score, and the PI score above 0.8 are considered significant. The PI value of the 11 predicted epitopes ranged from 0.809 to 0.911 and the epitopes with higher scores indicated greater solvent accessibility. Conformational epitopes and their associated parameters and scores revealed that epitopes with highest number of residues (110) were present in conformational epitopes from S protein and the minimum number of residues (3) was predicted from M protein (Table S8). The three dimensional structure and location of the conformational epitopes were displayed by ElliPro (Fig. 4).

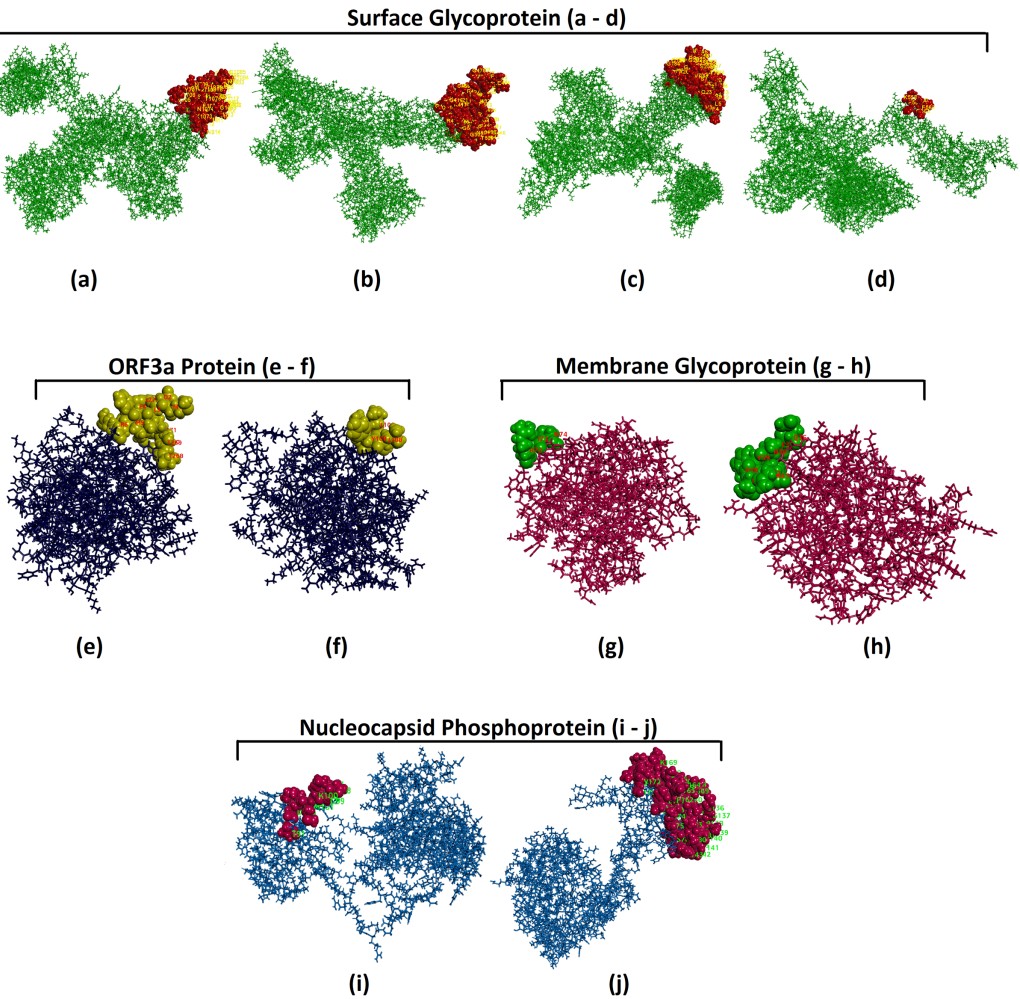

**Figure 4 Three-dimensional representation of B cell conformational epitopes.** Three-dimensional representation of B cell conformational epitopes of the structural proteins of SARS-CoV-2. The epitopes are represented by 3D structure, and the bulk of the protein is represented by sticks. (A–D) Images are of the surface glycoprotein, (E and F) images are of ORF3a Protein, (G and H) images are of membrane glycoprotein and (I and J) images are of Nucleocapsid Phosphoprotein.

## Analysis of physicochemical properties

Physicochemical properties of the SARS-CoV-2 structural proteins are described in Table S9. The values revealed that the S, ORF3a and ORF6 proteins were naturally acidic whereas the E, M, N protein were naturally basic. All the six structural proteins from SARS-CoV-2 used in the study had estimated half-life of 30 h in mammalian reticulocytes under in vitro conditions, whereas in yeast the estimated half life was more than 20 h. The least survival time of more than 10 h was estimated in *Escherichia coli*. Unlike the proteins, proposed MHC I epitopes had different half life. An estimated half life of less than an hour in mammalian reticulocytes was associated with QLTPTWRVY epitope derived from S protein, whereas maximum half life of 30 h was estimated for GVYFASTEK. On the other hand least estimated half life in yeast system was predicted for RFLYIIKLI

and however maximum estimated half life of more than 20 h was found in case of four epitopes (Table S9). Lastly in the most commonly used protein expression system that is, *E. coli* five epitopes had a life of more than 10 h.

Similarly, out of the four potential MHC II epitopes two had the maximum estimated half life of 20 h in mammalian reticulocytes, while, three epitopes had an estimated half life of more than 10 h in *E. coli*. In contrast, no MHC II epitope had an estimated half life of more than 30 min in yeast system (Table S9). These estimated half lives of MHC I and MHC II peptide epitopes suggested that most of the promising vaccine candidates could safely be produced in one or the other protein expression systems mentioned above.

## DISCUSSION

The advancing pandemic of corona virus disease 2019 (COVID-19) has resulted in death of more than 445,000 human population globally (*World Health Organization, 2020*). The disease is generated by severe acute respiratory syndrome corona virus 2 (SARS-CoV-2) (*World Health Organization, 2020*). Keeping the SARS-CoV-2 (RNA-virus) mutability in mind (*Twiddy, Holmes & Rambaut, 2003*; *Manzin et al., 1998*), a comprehensive vaccine needs to be designed to overcome the adverse effects of this viral infection. However, an efficacious vaccine development and mass production are expensive and can take several years to be completed. Therefore, an attempt was made to design a peptide-based vaccine using the immuno-informatics approaches to minimize the time required for searching a potent vaccine candidate for SARS-CoV-2. At present, distinct Bioinformatics approaches are available for the design and development of successful and safe new-generation vaccines (*María et al., 2017*; *Seib, Zhao & Rappuoli, 2012*). The advancement in computational immunology and newer immuno-informatics tools have created a broader way in developing the vaccine or vaccine candidates by the adequate understanding of the human immune response against a pathogen within a short period of time (*De Groot & Rappuoli, 2004*; *Korber, LaBute & Yusim, 2006*; *Purcell, McCluskey & Rossjohn, 2007*). The scheme of an epitope-based vaccine against rhinovirus, (*Lapelosa et al., 2009*) dengue virus, (*Chakraborty et al., 2010*) chikungunya virus, (*Islam, Sakib & Zaman, 2012*) Saint Louis encephalitis virus, (*Hasan, Hossain & Alam, 2013*) etc. has already been proposed.

In the present study, we first attempted to identify the potential vaccine candidates based on the T cell peptide epitope. In contrast to earlier vaccines, which are predominantly based on B cell immunity, vaccine based on T cell epitope has also been recommended as the host can induce a strong immune response by CD8[+] T cell against the infected cell (*Van Regenmortel, 2001*). Due to antigenic drift, any foreign particle can escape the antibody memory response mounted by B cells; however, the immune response generated by T cells usually provides long-lasting immunity. There are various specifications that need to be fulfilled by a peptide vaccine candidate. The potential epitopes proposed in our study satisfied all the criteria evaluated using computational tools.

The T-cell epitope was identified based on high threshold values (1.25) obtained in the output of NetCTL 1.2 tool. Primarily, more than one hundred fifty epitopes from six structural proteins were identified by selecting twelve super types of MHC-1 alleles. The antigenicity, immunogenicity and conservancy of the epitopes are considered as

important determinants. Therefore, by maintaining critical thresholds of the antigenicity, immunogenicity and conservancy of the epitopes, we picked thirty eight epitopes from structural proteins of SARS-CoV-2 (Table 1). These selected T-cell epitopes had a higher conservancy between 65.0% and 100.0%, which further support the feasibility of these predicted epitopes and indicate them as a potential vaccine candidate.

Most of the present day vaccines activate the immune system into allergic state (*McKeever et al., 2004*) by inducing type 2 T helper T (Th2) cells and immunoglobulin E (IgE). Consequently, allergenic property is one of the major hurdles in vaccine development. Hence, all the selected T-cell epitopes were screened for allergenicity by two computational tools; AllerTOP v.2.0 and AllergenFP 1.0. Altogether only eleven epitopes were classified by both the tools as non-allergens. Those eleven epitopes with all the characteristics of good vaccine candidates may be considered most important epitope in comparison with the other epitopes.

Another important factor in the selection of a potential vaccine is population coverage. The human leukocyte antigen alleles are remarkably polymorphic in diverse ethnic populations. Consequently, allele specificity of T-cell epitopes is considered as the initial criterion for the induction of proper immune responses in numerous ethnic human populations (*Stern & Wiley, 1994*). For all the eleven promising T cell vaccine candidates, the cumulative percentage of population coverage was measured. Overall the recommended epitopes from surface glycoprotein showed world population coverage of 80.37% followed by nucleocapsid phosphoprotein and ORF3a epitopes showing 68.10% and 54.43% of world population coverage, respectively (Table S4). The SARS-CoV-2 outbreak has resulted pandemic in which cases have been reported in almost all the countries of world (*World Health Organization, 2020*), so a vaccine candidate which can protect the majority of world's population is required.

However, the epitopes from membrane protein could cover only 31.04% population of World. Notably, the epitopes from the surface protein had population coverage of 89.08% for China, where the virus originated (*Zhou et al., 2020*) and 88.99% for Southeast Asia. In the list of badly hit countries, majority is from Europe (*World Health Organization, 2020*) and the epitope from S protein had coverage of 80.69% of Europe's population. The population coverage of 77.72% was obtained for USA where the highest number of cases has been reported (*World Health Organization, 2020*). The nucleoprotein epitopes covered 76.28% of Europe population followed by 69.53% and 63.26% of Italy and United States populations; respectively. Next, the ORF3a epitopes had 64.57% coverage of China's population followed by 63.85% and 57.98% of Hong Kong and Europe's population, respectively. Taken together all the suggested epitopes having higher population coverage may be considered as strong vaccine candidates.

The proper binding of the T cell epitope to the MHC I antigen binding cleft is essential for the induction of desired immune response (*Stern & Wiley, 1994*). The legitimate binding should result in a negative HADDOCK and *Z* scores. Thus the 3D structure of the proposed vaccine candidate was designed using PEP-FOLD and the crystal structure of the selected MHC allele was obtained from Protein Data Bank (PDB). Thereafter, removal of water and retrieval of chain wise structure of MHC alleles were performed using PyMol.
In the next step molecular docking simulation was executed with selected chain of the MHC as protein molecule and the proposed vaccine candidate as ligand using HADDOCK 2.4. The HADDOCK and $Z$ scores, the two most significant parameters in the results of HADDOCK indicated the predicted epitopes to be reasonable. The 3D and 2D interaction maps were derived using the HADDOCK best cluster model generated in result page by applying appropriate bio-informatics resources like PYMOL and Discovery studio. These structures exhibited appropriate binding of predicted epitopes in MHC I peptide binding cleft suggesting the pertinent selection of bio-informatics approach for epitope identifications. The re-docking and validation of docking method was carried out by using the seven crystal structures of MHC I and MHC II alleles and their corresponding peptide epitope obtained from PDB. The structures of both HLA allele and corresponding peptides were obtained using PyMol and re-docking was performed using HADDOCK 2.4. The docking procedures were same as that for the SARS-CoV-2 predicted epitopes. HADDOCK and $Z$ scores were in the acceptable range (negative values) and the 3D and 2D interaction results were similar to the corresponding PDB structures. Furthermore the results of HADDOCK re-docking were similar to those achieved by dockings performed using the predicted SARS-CoV-2 MHC I and MHC II epitopes, which reflected valid docking methodologies adopted in the present study. The physicochemical properties of proposed epitopes indicated that these can be produced in any three of the systems used for the expression of peptides, viz, mammalian cells, yeast cell or *E. coli* (Table S9)

Most of the current day vaccines are based on the B cell epitopes (*Sarkander, Hojyo & Tokoyoda, 2016*). BepiPred-2.0 might be viewed as the prime and most up-to-date B-cell epitope prediction computational tool as it exhibits notably good performance on both epitope data obtained from a vast number of linear epitopes taken from the IEDB database and on structural data of epitope derived from crystallography studies. LBtope is other robust tool for linear B-cell epitope prediction. It has been generated based on the experimentally proven non B-cell epitopes derived from the IEDB database. Antigenicity, allergenicity, toxicity and conservancy of the predicted B cell linear epitopes are prime determinants for identifying potential vaccine candidates. Therefore, all the four criteria were evaluated using different standard bioinformatics tools and potential epitopes were selected on the basis of high threshold values as fixed for T cell epitopes. Thus, on the basis of above criteria and conservancy altogether seven B cell epitopes from structural proteins were proposed as potential B cell vaccine candidates. The majority of the B-cell epitopes are discontinuous or conformational epitopes, and the quantum of this epitope is more than 90% (*Van Regenmortel, 2001*). Therefore, discontinuous B-cell epitopes were identified using ElliPro, a strong tool for the identification of conformational B cell epitopes. The tool identified four epitopes from surface glycoprotein followed by two epitopes each from the orf3a protein, membrane protein and nucleocapsid phosphoprotein. The extensive range of these conformational epitopes drawn on different proteins of SARS-CoV-2 indicated their potential as conformational B cell vaccine candidates.

An earlier study has reported a single epitope from spike protein having the conservancy of about 64% (*Oany, Emran & Jyoti, 2014*). Here we have reported several epitopes as potential vaccine candidates from five structural proteins of SARS-CoV-2.

As all the vaccine candidates need to be verified in clinical trials, the normal path of vaccine development, we propose the identified potential vaccine candidates should be pursued in clinical trials.

## CONCLUSIONS

In view of the present COVID-19 pandemic, for development of vaccine efficiently and within minimal time, vaccine candidates need to be identified at the early. Based on advanced computational approaches, we have altogether identified eleven potential T-cell epitopes, seven B cell linear epitopes and ten B cell conformational epitopes from the six structural proteins of SARS-CoV-2. Taken together these numerous potential vaccine candidates may provide important timely avenues for effective vaccine development against SARS-CoV-2. The future efforts may focus on the clinical trials of the multi-epitope vaccine candidates based on the present study.

## ACKNOWLEDGEMENTS

The authors acknowledge the infrastructure developed under the DBT-BUILDER Project by DBT.

### Funding

The authors received no funding for this work.

### Competing Interests

The authors declare that they have no competing interests.

### Author Contributions

- Rajesh Anand conceived and designed the experiments, performed the experiments, analyzed the data, prepared figures and/or tables, authored or reviewed drafts of the paper, and approved the final draft.
- Subham Biswal performed the experiments, analyzed the data, prepared figures and/or tables, authored or reviewed drafts of the paper, and approved the final draft.
- Renu Bhatt analyzed the data, authored or reviewed drafts of the paper, and approved the final draft.
- Bhupendra N. Tiwary analyzed the data, authored or reviewed drafts of the paper, and approved the final draft.

### Data Availability

The raw data is available in the Supplemental Files.

### Supplemental Information

Supplemental information for this article can be found online at http://dx.doi.org/10.7717/peerj.9855#supplemental-information.

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
