# Peer review of "Computational perspectives revealed prospective vaccine candidates from five structural proteins of novel SARS corona virus 2019 (SARS-CoV-2)"

_PeerJ, doi:10.7717/peerj.9855_

## Round 0.1 · original submission · Major Revisions

Three specialists in the field have evaluated this manuscript. They all have concerns related to this submission. One of the reviewers highlighted the need for further validation of the docking protocols. Considering the analysis carried out by the reviewers, I recommend a major revision in the paper.

Reviewer 1 ·

Basic reporting

The main idea of the work is to propose epitopes from the SARS-CoV-2 which can act as vaccines, using viral proteins, using computational methods.
The English language is well written and clear in all the manuscript. The authors addressed the main problem and ideas in the introduction section, as well as used appropriate references. On the other hand, I have some questioning described below:

• Lines 49 – 51: Please update WHO numbers.
• Lines: 79 – 80: Do you have any information about what is this variation? Please include some paper talking about it.
• Lines 87 – 91: Could you better describe ACE2/RBD importance in SARS infection here? I have bee reading many papers targeting its importance for viral pathogenicity, as well as drug development. Dong et al. (2020) just hypothesize about ACE2 role in viral infection, and I think you need more examples to confirm this sentence. Please complete your sentence with other references.

Figures 1, 5 and tables are well-representatives in most of them, except in the cases below:

-Figures 2 and 3: should be completed with 2D interaction maps and re-docking experiments.
-Figure 4: due to bad resolution I can't read coverage results.

Computational predictions of possible epitope activity were validated by docking studies with crystallographic structures from PDB. In this case, I found some flaws that are described in the experimental design.

I will recommend the paper publication only with major revisions, that will be described in the next sections.

Experimental design

Comments about experimental design are described below:

• Lines 126 - 128: VaxiJen server uses physicochemical proprieties in its analysis, instead of peptide alignment. Did you consider using another tool with alignment mode? Why VaxiJen is a better choice? Please, include an advantage to use it. Additionally, indicate here in methods the threshold that you considered.
• Lines 143 – 144: Please include the default parameters you used.
• Lines 162 – 163: Please include a reference for this statement.
• Lines 168 – 173: Too simple description of the determination of population coverage. Please, include more details about this part of the methods.
• Lines 185 -188: Too simple method description.
• Lines 189 – 195: What criteria did you use to select the best epitopes for modeling? What comes first in those characteristics that you considered (antigenicity, conservancy, etc.). Please, be clearer on what was more important to select your best epitopes.
• Lines 196 – 202: Is the Autodock Vina the best option for peptide docking? I think there is a problem considering this program instead of other peptide docking programs (Cluspro, Haddock, DockThor, etc.) and/or Autodock 4 as well, giving better evaluation results. In the case of Autodock Vina, its algorithm is more restricted for huge molecule evaluation, as well as its results, could not reflect several characteristics in peptide interaction, such as protonation states of the amino acids, charges and solvent accessibility for example. In this case, this could be a problem to represent the better pose of the molecule inside the protein of interest. I suggest you choose another program that uses a peptide docking approach, in order to validate this part of your work.
• Line 202: What Pymol version? Cite Pymol.
• Line 205: in Autodock Vina the program considers Affinity energy, which is a little different from binding energy. Change the term.
• Line 207: Setting up the grid box as default is not the best choice. Depending on the type of binding site that you have, it will require smaller or bigger boxes.
• Lines: 227 – 230: Wrong terminology. I think the authors wanted to say physicochemical instead.
I would like to do some considerations/questioning about the methods section:

1) The authors used complete or partial protein sequences? Please include this information in the methods section.
2) As you are proposing a vaccine using protein sequences for computational predictions, I strongly recommend that you include variability analysis using sequences from different viral strains. From what region of the world the sequences that you used came from? They are consensus sequences from different countries or from just one country?
3) I recommend you include a sequence alignment for each protein group, or you can align just the region that you used as epitopes for each protein type

Validity of the findings

This work presented important findings of new peptide sequences that can be used for vaccine development. On the other hand, there are computational validations and other explanations that must be included. The study didn't say how many sequences for each protein type they used, and from where these sequences come from so is very hard to accept that a global vaccine could be developed using these findings. Furthermore, the results and consequently discussion sections lack some information about best epitope selection, as well as docking results and validation. I suggest some changes below.

Lines 256 – 257: what criteria the authors used to put a 250 nM IC50 cutoff? Please, explain this in methods or in results/discussion sections.
Line 260: What is this score of 65%? I didn’t find this in methods. Include for what reason you used this.
Lines 329 – 354: Docking results section needs to be re-written with more details. There is no validation for docking results, and there is no 2D interaction maps peptide-protein. Please include the 2D interaction maps for each docking calculation. I recommend including an epitope re-docking for validation: 3C9N, 5IEK and 5VGD crystals are protein-peptide complexes. I should use some of them for re-docking and validation of your docking methods.
Lines 434 – 435: in Physicochemical properties analysis what you consider as stable structures?

Reviewer 2 ·

Basic reporting

The manuscript has a clear and professional use of English.
Although the COVID-19 literature increases by the day, and there are many preprints, the references used are adequate for the methodology applied.
The paper has good structure, the protein models are properly drawn and colored, although figure 4 is impossible to read. Maybe they can be included at full size as supplemental material and just make a reference to the important alleles in results and discussion.
In figure 5, it would be nice if some sequence numbering is included. Also, it is not clear if the images correspond to different proteins or are views of the same protein. Please explain in more detail.

In line 225. and 227, it should be “physicochemical” instead of “Physiological”, probably a typo.
In the same section please also include the prediction of stability if expressed on E. coli

Maybe design the figure using a “surface glycoprotein” label, and then include figures a-d. Then a label “ORF3a protein” and the corresponding g-h images and “membrane glycoprotein”, and a fourth label “nucleocapsid phosphoprotein” for i-j images.

Experimental design

The methods are described in detail and the parameters used in each algorithm are included.
I could not find details of the program used to display the structures, please include the reference and the PDB access for each structure (if taken from there) or if these are your predictions, how they were modeled.

Validity of the findings

Novel work that predicts potential targets towards a COVID-19 vaccine. The pipeline is clear and it helps the non-specialist to understand the procedures. Table 2, although detailed, is great because it resumes all the scores that point out towards the target peptides that are worth further investigation.

I consider important, since physicochemical properties are reported, that the author proposes the challenges of expressing these peptides as vaccines, Which ones would be more suitable for bacterial expression, pros, and contras, and also if there are issues with glycosylation. What are the more adequate systems for expressing and purifying the peptides?

Labeling in figure 2 could be done with 1-letter amino acid symbols to reduce the crowding of the stick figures. Letters in figure 2c are too small, in 2F hard to read, in 2i too big. In all of them, a one-letter amino acid code would be better.

Finally, I suggest that the author takes a final review of the current literature since there are new reports by the day, of the current status of vaccine development.

Additional comments

The paper is novel and understandable, please keep it updated so it stays as an important resource in the avalanche of COVID-19 publications.

·

Basic reporting

Comments for the author on English.
Figure 4 - text in figure is far too small, impossible to read, it might need a key.

Experimental design

No comment.

Validity of the findings

No comment.

Additional comments

The introduction is highly informative and the literature is well referenced and appropriate. It provides good background and context to the reporting of the study. There are numerous minor grammatical errors, in addition to those I have identified in my comments, so I would recommend careful and thorough proofreading. The research question is well defined and very timely. The knowledge gap is identified clearly, as is the approach to the intended filling of the gap.

The study addresses the important question of identifying effective epitopes for SARS-CoV-2 by the coherent application of a number of well established bioinformatics methods that predict antigenicity, immune protein epitopes, immunogenicity, allergenicity and toxicity, in this case applied to viral protein sequences, which when taken together allow cross-referencing of their results to address the specific question concerning epitope-based vaccine design. The methods are appropriate to the problem and the authors should be commended on a thorough and well-directed study. However, some clarifications of the core analytical approach and extensive grammatical corrections are required.

Line by line comments:

"anticipation" should be replaced by "prediction". This word is frequently misused and should be corrected throughout.
Line 146 - missing "For" the recognition...
Line 147 - was - were. This is also a frequent error
Line 150 - 250? The units need to be included, nM?
Line 151 - "Lower IC50 value signifies higher binding" - this should be qualified or prefaced by the term "In general, for similar ligands," as the maximal bound levels of different ligands can vary considerably even at the same site, meaning that comparison of IC50 values for different ligands in bringing about 50% inhibition only relates to the required concentration of that ligand relative to that for the highest attainable inhibition for that ligand. It doesn't allow for comparison of the level of binding of different ligands, or actual concentration of ligand at a given site.
Line 151 - ingenious - not clear what is meant here.
Line 155 - combined score - how is this derived?
Line 156 - performed - made
Line 183 - created - developed
Line 186 - was - were
Line 187 - The - An
Line 190 - was - were
Line 194 - implied - implicated, or better still, taken forward
Line 207 - Eventually - remove this
Line 217 - generated - developed
Line 222 - demonstrate - provides
Line 226 - upper case F in For, should be lower case
Line 240 - the - not needed
Line 249 - the (antigenicity) - not needed
Line 251 - Surprisingly? Why is this a surprise?
Line 251 - Similarly? - the values are very different
Line 252 - immunogenicity value - immunogenicity values
Line 253/254 - Table S2 details 82 epitopes rather than 83. The combined scores are shown in the table. However, it is not clear how these scores are derived. Further, the selection criteria for the 82 epitopes for further study is not described, this requires clarification, particularly if VaxiJen indicated only 49 epitopes to be antigenic. This contradicts the assertion in line 155/156 - "the final selection for further study was performed after the (prediction) of antigenicity by VaxiJen v2.0 server....", and suggests that a proportion of the negative results obtained using VaxiJen, which represent an important observation, are being overlooked.
Line 257 - was - were; IC50 values
Line 267 - allergic - allergenic; allergen - allergenic
Line 268 - was - were
Line 289 - fetched - gathered
Line 291 - should read - although E protein epitope “LLFLAFVVF” exhibited binding....
Line 298 - non-allergen - non-allergenic
Line 299 - candidates
Line 308 - two other epitopes...
Line 318-320 - ....ORF3a was found to be present.... (the two sentences could be condensed into one)
Line 321 - sentence is not clear at all, should be rephrased
Line 324 - resulted - result
Line 330 - considered to be potential...
Line 347 - As a representative of what?
Line 356 - amid - across
Line 369 - was - were
Line 384 - candidates
Line 385 - adjective missing before "antigenicity"
Line 387 - the spike protein
Line 396 - values
Line 400 - candidates
Line 415 - remove "only"
Line 430 - exhibited - displayed
Line 444 - design a peptide-based
Line 460 - Cells - cells (lower case)
Line 468 - thresholds
Line 470/471 - further ensure the acceptance - further supports the feasibility
Line 472 - incomplete sentence - it looks like it should be part of the previous sentence?
Line 478 - considered as the most important epitopes, no need for "rest of the"
Line 482 - criterion
Line 485-488 - reference should be made to Table-S4
Line 488 - The SARS-CoV-2 outbreak has resulted...
Line 505 - the selected
Line 506 - simplified?
Line 507 - no need for "investigation"
Line 514 - most up-to-date
Line 515 - exhibits notably good performance
Line 519-521 - incomplete sentence
Line 525 - The majority
Line 527 - the ElliPro - "the" not needed
Line 532 - An earlier study
Line 535-536 - we propose the identified potential vaccine candidates should be pursued in clinical trials.
Line 539 - at the earliest - early
Line 542 - vaccines - vaccine
Line 543 - directives - directions / avenues; no need for "the"

Figure 1 legend - proteins
Throughout Figure 2 and Figure 3 legends - odd use of "as"
Figure 4 legend - proteins
Figure 4 - text in figure is far too small, impossible to read, it might need a key.

Otherwise, the figures are very good, relevant and the legends are informative.

The tables are fine. Table 4 should perhaps be moved to the supplementary.

There are no legends provided for the supplementary tables.

In all, this is a good manuscript, in my view very worthy of publication, which successfully applies and integrates a number of bioinformatics tools and resources in a coherent, meaningful way to address the challenge of the day. The basis of some scores and selection or filtering processes requires clarification and the manuscript needs busy checking of the English to ensure the widest possible understanding of the findings.

---

## Round 0.2 · accepted · Accept

Thank you for your careful revision and appropriate responses to the previous critiques. I have sent your article back to two previous reviewers. These reviewers and I are now satisfied with the revisions and feel your paper is nearly ready for publication. The editorial office will be in touch with further details and some additional adjustments.

Reviewer 1 ·

Basic reporting

no comment

Experimental design

no comment

Validity of the findings

no comment

Additional comments

The authors adjusted the manuscript as recommended. I would like to thank you for agreeing with my suggestions for improving the work. The manuscript is now suitable for publication.

·

Basic reporting

The manuscript is much improved. The authors have responded positively to the key suggested revisions and comments. The English too is improved, but could be bettered further by a final general proofreading, though I don't think that the remaining shortcomings impact upon the understanding or reproducibility of the work. This is ultimately an editorial decision.

A few of the more obvious instances that can be quickly amended:

Line 363 : recognized - change to identified
Line 530 : Lower case Haddock, upper case everywhere else
Line 572-574 : The sentence doesn't make its point
Line 659 / 669 : Species name should be in italics
Line 769 : yeast

Experimental design

No further comments.

Validity of the findings

No further comments.

Additional comments

This is a good and timely study and a solid foundation for further work.